# Cannabinoid combination targets *NOTCH1*-mutated T-cell acute lymphoblastic leukemia through the integrated stress response pathway

Elazar Besser, Anat Gelfand, Shiri Procaccia, Paula Berman, David Meiri*

The Laboratory of Cancer Biology and Cannabinoid Research, Faculty of Biology, Technion – Israel Institute of Technology, Haifa, Israel

**Abstract** In T-cell acute lymphoblastic leukemia (T-ALL), more than 50% of cases display autoactivation of Notch1 signaling, leading to oncogenic transformation. We have previously identified a specific chemovar of Cannabis that induces apoptosis by preventing Notch1 maturation in leukemia cells. Here, we isolated three cannabinoids from this chemovar that synergistically mimic the effects of the whole extract. Two were previously known, cannabidiol (CBD) and cannabidivarin (CBDV), whereas the third cannabinoid, which we termed 331-18A, was identified and fully characterized in this study. We demonstrated that these cannabinoids act through cannabinoid receptor type 2 and TRPV1 to activate the integrated stress response pathway by depleting intracellular $Ca^{2+}$. This is followed by increased mRNA and protein expression of ATF4, CHOP, and CHAC1, which is hindered by inhibiting the upstream initiation factor eIF2α. The increased abundance of CHAC1 prevents Notch1 maturation, thereby reducing the levels of the active Notch1 intracellular domain, and consequently decreasing cell viability and increasing apoptosis. Treatment with the three isolated molecules resulted in reduced tumor size and weight in vivo and slowed leukemia progression in mice models. Altogether, this study elucidated the mechanism of action of three distinct cannabinoids in modulating the Notch1 pathway, and constitutes an important step in the establishment of a new therapy for treating *NOTCH1*-mutated diseases and cancers such as T-ALL.

**\*For correspondence:**
dmeiri@technion.ac.il

## eLife assessment

This **important** study follows up on previous work defining the anti-leukemic effects of a previously characterized cannabis extract on Notch-activated T cells and identifies several pathways that mediate its anti-cancer activity including the ER calcium and integrated stress response. The evidence is **solid**, but several concerns remain including the over reliance on a single cell line for the majority of the studies and lack of integration of the observations with existing literature

## Introduction

T-cell acute lymphoblastic leukemia (T-ALL) is an aggressive cancer, characterized by immature T-lymphoblasts (*Girardi et al., 2017*; *Vadillo et al., 2018*). In over 50% of cases, activating mutations in *NOTCH1* were identified, generating much interest in targeting the Notch1 signaling pathway in this disease (*Sorrentino et al., 2019*; *Ferrando, 2009*; *McCarter et al., 2018*). The Notch1 protein is a single-pass transmembrane receptor that is expressed on the plasma membrane. It regulates evolutionarily conserved signaling that controls developmental processes, cell fate determination,

and tissue homeostasis. It is a member of the Notch receptor family, which consists of four receptors (Notch1-4) in humans (*Ferrando, 2009*; *Fang-Fang et al., 2021*).

The Notch1 receptor-signaling pathway is activated through a series of proteolytic cleavages at three different sites (S1–S3). S1 cleavage is carried out by a Furin-like convertase in the trans-Golgi apparatus, resulting in a heterodimeric receptor that is transported to the membrane where it can bind ligands from a neighboring cell. Upon binding of a ligand, cleavage is performed at the next site and the activation of the Notch1 receptor-signaling pathway is initiated, enabling the cleavage of S3 in the transmembrane region that causes the release of the Notch1 intracellular domain (NICD), which translocates to the nucleus where it promotes transcription of target genes involved in cell growth (*Katoh and Katoh, 2020*).

Aberrant Notch1 activity is associated with many other leukemia types and other cancers, including prostate, lung, ovarian, and renal (*Qiu et al., 2018*; *Rice et al., 2019*; *Li et al., 2018*; *Kahn et al., 2018*; *Vinson et al., 2016*). A new and promising emerging field in cancer treatment is medical Cannabis and its unique active compounds, the phytocannabinoids. Phytocannabinoids affect the body by means of the endocannabinoid system (eCBS), a ubiquitous neuromodulatory signaling system that has widespread functions throughout the body. Notable eCBS receptors include cannabinoid receptor type 1 (CB1) and CB2, which belong to the family of G protein-coupled receptors, and transient receptor potential vanilloid type 1 (TRPV1), which is an ion channel (*Punzo et al., 2018*). The eCBS participates in many physiological activities, including the regulation of $Ca^{2+}$ homeostasis (*Jeon et al., 2023*; *Muller et al., 2018*; *Laguerre et al., 2021*).

Medical Cannabis is already commonly used by cancer patients for its palliative effects, it was shown to stimulate appetite while alleviating symptoms such as nausea, vomiting, and pain (*Abrams and Guzman, 2015*). However, accumulating evidence demonstrated phytocannabinoids directly affect tumor development in cell lines and animal models by modulating key cell-signaling pathways (*Abrams, 2016*; *Duran et al., 2010*; *Pagano et al., 2021*). Studies demonstrated that purified phytocannabinoids can inhibit proliferation, metastasis, and angiogenesis and exert pro-apoptotic effects in a variety of cancer cell types such as lung, breast, prostate, skin, intestine, glioma, lymphoma, pancreas, and uterine cancers (*Kovalchuk and Kovalchuk, 2020*; *O'Reilly et al., 2022*; *Blázquez et al., 2004*). In preclinical models, treatment with phytocannabinoids led to tumor regression of different cancer types (*McAllister et al., 2015*; *Hinz and Ramer, 2022*).

Concerning T-ALL, some studies using phytocannabinoids and their respective formulations (such as dronabinol) have shown anti-proliferative as well as pro-apoptotic effects in selected leukemia cell lines and native leukemia blasts cultured ex vivo (*Kampa-Schittenhelm et al., 2016*; *Scott et al., 2017*; *Olivas-Aguirre et al., 2019*). Nevertheless, the mechanisms of phytocannabinoid-mediated antitumor effects are not yet fully understood. Most of the studies that examined the anti-cancer effects of Cannabis have focused mainly on the two major phytocannabinoids (−)-*trans*-Δ⁹-tetrahydro cannabinol (Δ⁹-THC) and cannabidiol (CBD), but these are just two of more than 140 different phytocannabinoids that have been identified in different Cannabis chemovars (*Berman et al., 2018*; *Hanuš et al., 2016*).

We have previously demonstrated that different Cannabis extracts with unique phytocannabinoid compositions impaired the survival and proliferation of specific cancer cell lines, suggesting that the effect of a particular Cannabis extract on a specific cancer cell line relies on its chemical composition (*Baram et al., 2019*). Following this framework, we successfully matched a specific CBD-rich Cannabis extract to T-ALL leukemia cells that harbor a *NOTCH1* mutation (*Besser et al., 2023*). In the present study, we investigated which specific metabolites in the whole extract are responsible for T-ALL elimination, and elucidated the molecular mechanism by which they mediate their antitumor effects in this type of cancer.

## Results

We have previously shown a specific CBD-rich Cannabis extract selectively induces apoptosis in T-ALL leukemia cell lines that harbor a *NOTCH1* mutation. Treatment led to accumulation of the full-length immature form of Notch1 in the membrane, correspondingly with reduced NICD protein expression and transcription activity (*Besser et al., 2023*).

## A combination of three phytocannabinoids induces apoptosis and reduces NICD expression

The CBD-rich extract was more potent than pure CBD or other high-CBD chemovars (*Besser et al., 2023*). This implied a specific combination of metabolites from the plant potentiate the full anti-tumor effect. Using semi-preparative HPLC, we separated the whole extract into four 10 min retention time fractions (*Figure 1A*). A full list of the phytocannabinoids constituting each fraction is presented in *Supplementary file 1*. We assessed the effect of each fraction on the viability of MOLT-4 cells (*Figure 1—figure supplement 1*) and found fraction 2 reduced viability to a similar extent as the whole extract. Further analysis revealed that out of the four fractions, only fraction 2 induced apoptotic cell death (*Figure 1B*), reduced the protein levels of the NICD (*Figure 1C and D*) and elevated the levels of cleaved caspase-3 (*Figure 1D*, *Figure 1—source data 1*). Thus, we excluded all the phytocannabinoids present in the other three fractions, such as $\Delta^9$-THC in fraction 3.

To pinpoint the active phytocannabinoids in fraction 2, we further separated this fraction into individual phytocannabinoids using centrifugal partition chromatography (CPC) followed by semi-preparative HPLC (C1-C5 in *Figure 1E*). We identified the specific compounds in the isolated peaks (*Supplementary file 2*), C2 was identified as cannabidivarin (CBDV) and C5 as CBD. C3 and C4 were identified as cannabidiolic acid (CBDA), the precursor to CBD, and as CBD-C4, respectively, but their concentrations were very low. C1 was identified as 331-18A, a phytocannabinoid first identified by our team in decarboxylated CBD-rich Cannabis chemovars (*Berman et al., 2018*), we characterized 331-18A using ESI-LC-MS/MS as having a deprotonated m/z of 331.2279 and MS/MS fragments resembling those of CBD with the addition of one hydroxyl group. To confirm the identification, the purity of the three isolated compounds was determined by UHPLC/UV (*Figure 1—figure supplement 2A–C*).

Here, we further elucidated the chemical structure of 331-18A by nuclear magnetic resonance (NMR) with CBD as the reference (*Figure 1—figure supplement 3* and *Supplementary file 3*). The NMR data of CBD was in close agreement with the literature (*Marchetti et al., 2019*). The spectra of CBD and 331-18A were very similar with the exception of the disappearance of the external double bond ($C_8$-$C_{10}$) and the appearance of a tertiary alcohol instead. The absolute stereochemistry of 331-18A was confirmed by specific rotation determination with a polarimeter in which 331-18A showed $[\alpha]^{20}_D$ −56.85 which is comparable with the $[\alpha]^{20}_D$ −64.16 of both the CBD that was isolated from the fraction (C5) and a commercial standard of CBD. These results suggested that similarly to CBD, the two substituents at positions 1 and 6 of the cyclohexene ring are in a *trans* configuration (*Figure 1F*).

To examine the cytotoxic properties of the isolated compounds and their combinations on MOLT-4 cells, the isolated phytocannabinoids were prepared in the same amounts and proportion as observed in the whole extract; e.g., if 1 µg of the whole extract contained 50% CBD, we used 0.5 µg of the purified CBD and 1 µg of the whole extract in the respective treatments. Compared to the whole extract, the phytocannabinoids 331-18A and CBDV alone did not induce apoptosis or had minor effects on the cells, and CBD alone induced only 55% cell death compared to the whole extract (*Figure 1G*). When CBD was combined with either 331-18A or CBDV, the percent of apoptotic cells increased to 58% and 61%, respectively. The combination of all three phytocannabinoids led to the greatest cytotoxic effect and only this combination was comparable to the whole extract (88.29% ± 4.31% and 93.85% ± 1.64%, respectively). We further evaluated the effect of the three phytocannabinoids separately or in different combinations on the protein expression of NICD (*Figure 1H and I*, *Figure 1—source data 2*) and found that all combinations of phytocannabinoids led to a significant reduction in NICD expression, but only when all the three were combined, NICD levels decreased to the same extent as with the whole extract.

## 331-18A, CBDV, and CBD mediate their effect through CB2 and TRPV1 to trigger Ca²⁺ flux

Phytocannabinoids usually operate through specific cannabinoid receptors in the eCBS. To elucidate which endocannabinoid receptors mediate the antitumor effect, we first evaluated in MOLT-4 cells the mRNA expression of *CNR1* and *CNR2* (for CB1 and CB2, respectively), as well as G protein-coupled receptor 55 (*GPR55*), *TRPV1*, transient receptor potential cation channel, subfamily A, member 1 (*TRPA1*), and transient receptor potential cation channel subfamily M (melastatin) member 8 (*TRPM8*) (*Figure 2A*). The $\Delta$CT results showed low expression for *CNR1*, *TRPA1,* and *TRPM8*, medium expression for *CNR2* and *GPR55*, and high expression for *TRPV1*. We tested antagonists for those receptors

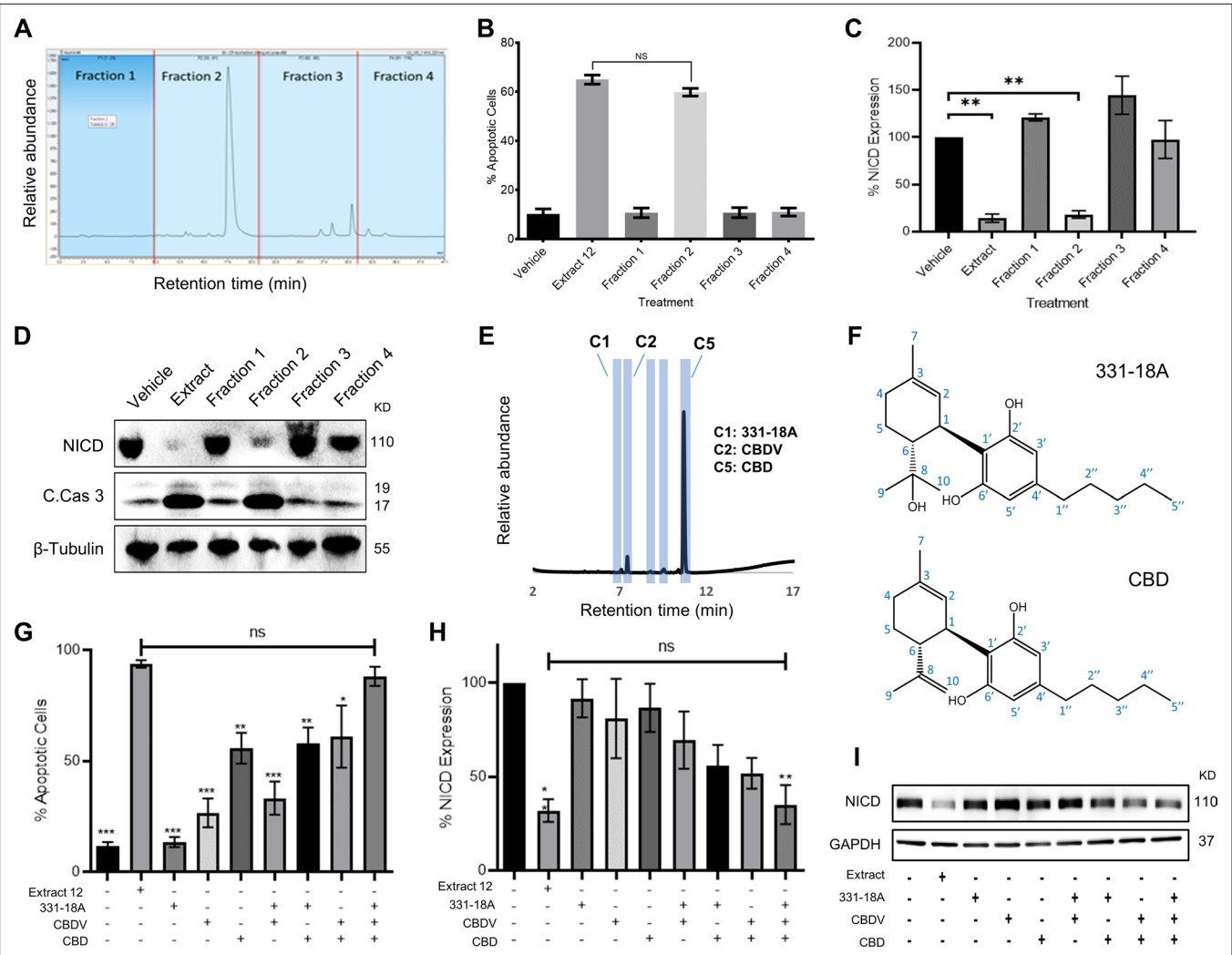

**Figure 1.** Identifying specific phytocannabinoids responsible for the cytotoxic effect of the whole extract. (**A**) A semi-preparative HPLC-UV chromatogram of the whole extract showing the four fractions collected every 10 min. (**B–D**) MOLT-4 cells were treated with either vehicle, whole extract, or fractions 1–4 (3 µg/mL) and (**B**) apoptosis was analyzed after 24 hr (N=3) via Annexin V/PI, (**C, D**) Notch1 intracellular domain (NICD) and cleaved caspase-3 (C. Cas. 3) expressions were evaluated (N=3) after 3 hr with β-tubulin as the loading control. A representative blot is shown. (**E**) UHPLC/ UV chromatogram of fraction 2, the specific compounds constituting the fraction are marked (C1–C5). (**F**) Chemical structure and peak assignment of cannabidiol (CBD) and 331-18A according to [1]H and [13]C nuclear magnetic resonance (NMR). Atom numbering is according to the monoterpene numbering system. (**G–I**) MOLT-4 cells were treated with the whole extract (3 µg/mL) or with 0.06 µg/mL 331-18A, 0.06 µg/mL cannabidivarin (CBDV), and 1.5 µg/mL CBD, their corresponding concentrations in the extract, and their different combinations. Cells were assessed for (**G**) apoptosis after 24 hr (N=3) via Annexin V/PI assay and (**H, I**) NICD expression after 3 hr (N=3) with GAPDH as the loading control. Results are presented as mean ± SEM and statistically analyzed with one-way ANOVA (*$p<0.05$, **$p<0.01$, ***$p<0.001$).

The online version of this article includes the following source data and figure supplement(s) for figure 1:

**Source data 1.** Original file for the western blot presented in *Figure 1D*.

**Source data 2.** Original file for the western blot presented in *Figure 1I*.

**Figure supplement 1.** Fraction 2 is as effective as the whole extract in reducing the viability of MOLT-4 cells.

**Figure supplement 2.** Specific identification of phytocannabinoids with anti-cancer properties by spectral matching and peak purity.

**Figure supplement 3.** Cannabidiol (CBD) and 331-18A nuclear magnetic resonance (NMR) spectra.

that had either high or medium expression: TRPV1 (AMG9810), CB2 (AM630 or SR-144,528), GPR55 (CID), and general Gq GPCR (BIM-46187) as a control, and evaluated whether they were able to rescue NICD expression upon treatment with the whole extract (*Figure 2—figure supplement 1A and B*, *Figure 2—figure supplement 1—source data 1*). Only the CB2 and TRPV1 antagonists rescued the

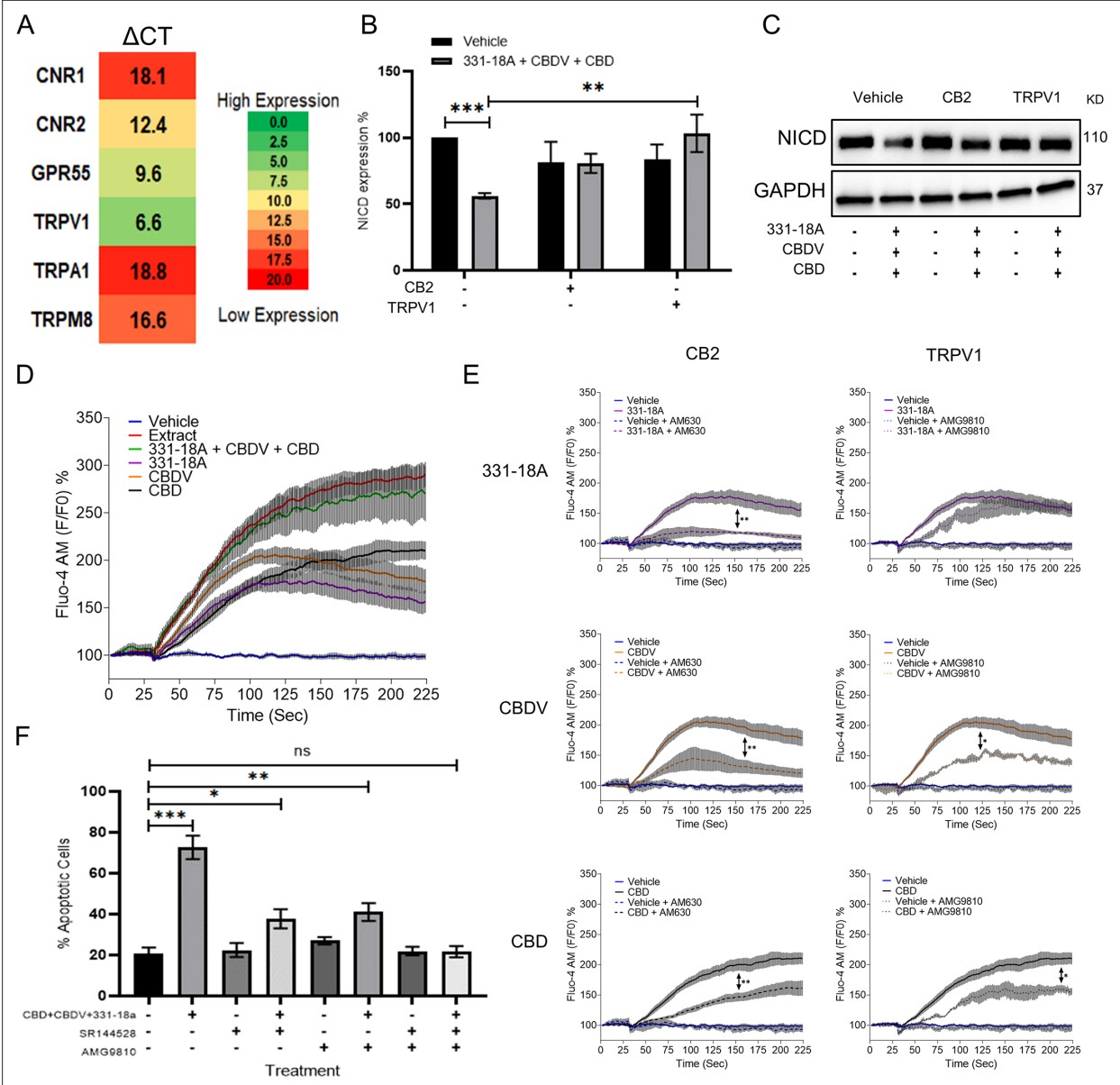

**Figure 2.** 331-18A, cannabidivarin (CBDV), and cannabidiol (CBD) mediate their effect through CB2 and TRPV1. (**A**) The mRNA levels of receptors *CNR1* (cannabinoid receptor type 1 [CB1]), *CNR2* (CB2), *GPR55*, *TRPV1*, *TRPA1*, and *TRPM8* were evaluated by qRT-PCR. Gene expression levels were calculated as ΔCT normalized to *GUSB* housekeeping gene. Results are presented as mean expression of three biological replicates. High ΔCT values indicate low receptor expression. (**B**) MOLT-4 cells were pretreated with 50 μM antagonist to CB2 (AM630) or TRPV1 (AMG9810) for 30 min, then treated with the combination of the three cannabinoids for 3 hr. Notch1 intracellular domain (NICD) expression was evaluated by western blot (N=3) with GAPDH as the loading control and statistically analyzed with an unpaired Student's t-test (**p<0.01, ***p<0.001). (**C**) Representative blots of B. (**D**) Calcium release by MOLT-4 cells was measured with the Fluo-4 calcium probe immediately following treatment with either vehicle, whole extract, a combination of 331-18A, CBDV, and CBD, and each cannabinoid separately. The calcium curves represent an average of three independent experiments. (**E**) Reduction of each phytocannabinoid-induced calcium release by pretreatment with 50 μM of antagonist to CB2 (AM630) or TRPV1 (AMG9810) for 30 min. The calcium curve for each compound is given again in this graph to compare the effects with the antagonists (N=3, two-way ANOVA, *p<0.05, **p<0.01). (**F**) MOLT-4 cells were pretreated with 50 μM antagonist to CB2 (SR-144,528) or TRPV1 (AMG9810) or both for 30 min, then treated with the combination of the three cannabinoids or left untreated. Apoptosis was analyzed after 24 hr (N=3) by Annexin V/PI. Results are presented as mean ± SEM and statistically analyzed with one-way ANOVA (*p<0.05, **p<0.01, ***p<0.001).

The online version of this article includes the following source data and figure supplement(s) for figure 2:

**Source data 1.** Original file for the western blot presented in *Figure 2C*.

**Figure supplement 1.** Notch1 downregulation by extract 12 is mediated through CB2 and TRPV1 followed by ATF4-CHOP-CHAC1 signaling pathway.

**Figure supplement 1—source data 1.** Original file for the western blot presented in *Figure 2—figure supplement 1B*.

expression of NICD upon treatment with the whole extract, therefore we tested whether they were able to rescue NICD expression after treatment with the combination of 331-18A, CBDV, and CBD (*Figure 2B and C*, *Figure 2—source data 1*). The CB2 antagonist rescued the expression from 56% to 87%, and the TRPV1 antagonist rescued NICD expression to 100% by itself.

TRPV1 is a $Ca^{2+}$ channel and eCBS receptors are known to trigger $Ca^{2+}$ flux (*Muller et al., 2018*; *Laguerre et al., 2021*). Therefore, as the next step we tested the effect of the three cannabinoids on cytosolic $Ca^{2+}$ levels. We used the Fluo-4 calcium probe to measure the effect of 331-18A, CBDV, and CBD combination on MOLT-4. Each of the molecules separately caused some elevation in $Ca^{2+}$, but only their combination had an effect on $Ca^{2+}$ to the same extent as the whole extract (*Figure 2D*). We tested $Ca^{2+}$ levels when cells were pretreated with the antagonists to CB2 and TRPV1. Both antagonists inhibited the elevation in response to the whole extract (*Figure 2—figure supplement 1C and D*). When 331-18A, CBDV, and CBD were tested separately with antagonists to either CB2 or TRPV1 (*Figure 2E*), CB2 but not TRPV1 antagonist diminished 331-18A induced $Ca^{2+}$ increase, indicating this cannabinoid exerts its effect through CB2, while both antagonists significantly decreased the $Ca^{2+}$ elevation by CBDV and CBD. Next, we assessed whether inhibition of the receptors rescues MOLT-4 cells from the cytotoxic properties of the cannabinoid combination (*Figure 2F*). Treatment with the inhibitors by themselves, separately or both together, did not induce MOLT-4 cell death. Upon treatment with the combination of 331-18A, CBDV, and CBD, inhibition of CB2 only partially rescued the cells from treatment-induced apoptosis, reducing the apoptotic rate from 75% of the combination-treated cells to 40%. However, the apoptotic rate was still significantly higher than that of the untreated control. Inhibition of TRPV1 only also partially rescued the cells, to ~50%, but again the apoptotic rate remained significantly increased relative to the untreated control. Inhibition of both CB2 and TRPV1 simultaneously rescued the cells from treatment-induced apoptosis, confirming the effect of the three cannabinoids is mediated through both receptors.

## Notch1 downregulation by cannabinoids is mediated through the integrated stress response pathway

To elucidate the mechanism by which the Notch1 pathway is downregulated following treatment, we used Affymetrix analysis to measure and compare the gene expression profile of MOLT-4 cells after 3 hr of treatment with either vehicle or the whole extract (*Figure 3A*). A list of the 10 most increased- and decreased-abundance genes following treatment is presented in *Supplementary file 4*. ChaC glutathione-specific γ-glutamylcyclotransferase 1 (*CHAC1*), a negative regulator of the Notch signaling pathway (*Chen et al., 2017*), was identified as the second most increased-abundance gene. CHAC1 acts by inhibiting the furin-like cleavage of Notch1, preventing the formation of the mature form of Notch1 (*Chi et al., 2012*), as we have previously demonstrated upon treatment with the whole extract (*Besser et al., 2023*). The gene *DDIT3* that encodes C/EBP homologous protein (CHOP) and activating transcription factor 4 (*ATF4*), ER-stress signaling pathway markers that are upstream to *CHAC1* (*Joo et al., 2015*), had also a significant increase in their abundance. We confirmed the Affymetrix results by qRT-PCR showing that the mRNA of *ATF4*, *DDIT3,* and *CHAC1* is indeed significantly increased after treatment with a combination of 331-18A, CBDV, and CBD (*Figure 3B–D*, respectively) or the whole extract (*Figure 3—figure supplement 1A–C*). A significant increase was apparent 60 min after treatment. Using specific antibodies to ATF4, CHOP, and CHAC1, we verified the protein expression level of all three was also significantly increased at 60 min after treatment (*Figure 3E–H*, *Figure 3—source data 1*). The protein expression levels of CHAC1 in MOLT-4 cells were also significantly increased upon treatment with the whole extract (*Figure 3—figure supplement 1D, E*, *Figure 3—figure supplement 1—source data 1*).

To further understand the mechanism, we evaluated how pretreatment with antagonists to the cannabinoid receptors through which 331-18A, CBDV, and CBD exert their effect, CB2 and TRPV1, affected *ATF4*, *DDIT3,* and *CHAC1* expression following treatment (*Figure 3I–K*). Inhibition of CB2 reduced the treatment-induced increase in mRNA expression, but a trend of increase for ATF4 and CHOP remained at 60 min, and CHAC1 expression was significantly increased. Inhibition of TRPV1 rescued from treatment-induced induction of all three targets. Similar effects were found when pretreatment with antagonists to CB2 and TRPV1 was followed by treatment with the whole extract (*Figure 3—figure supplement 1F-H*); and the increase in the expression levels of CHAC1 protein was abolished when MOLT-4 cells were treated by either antagonist (*Figure 3—figure supplement*

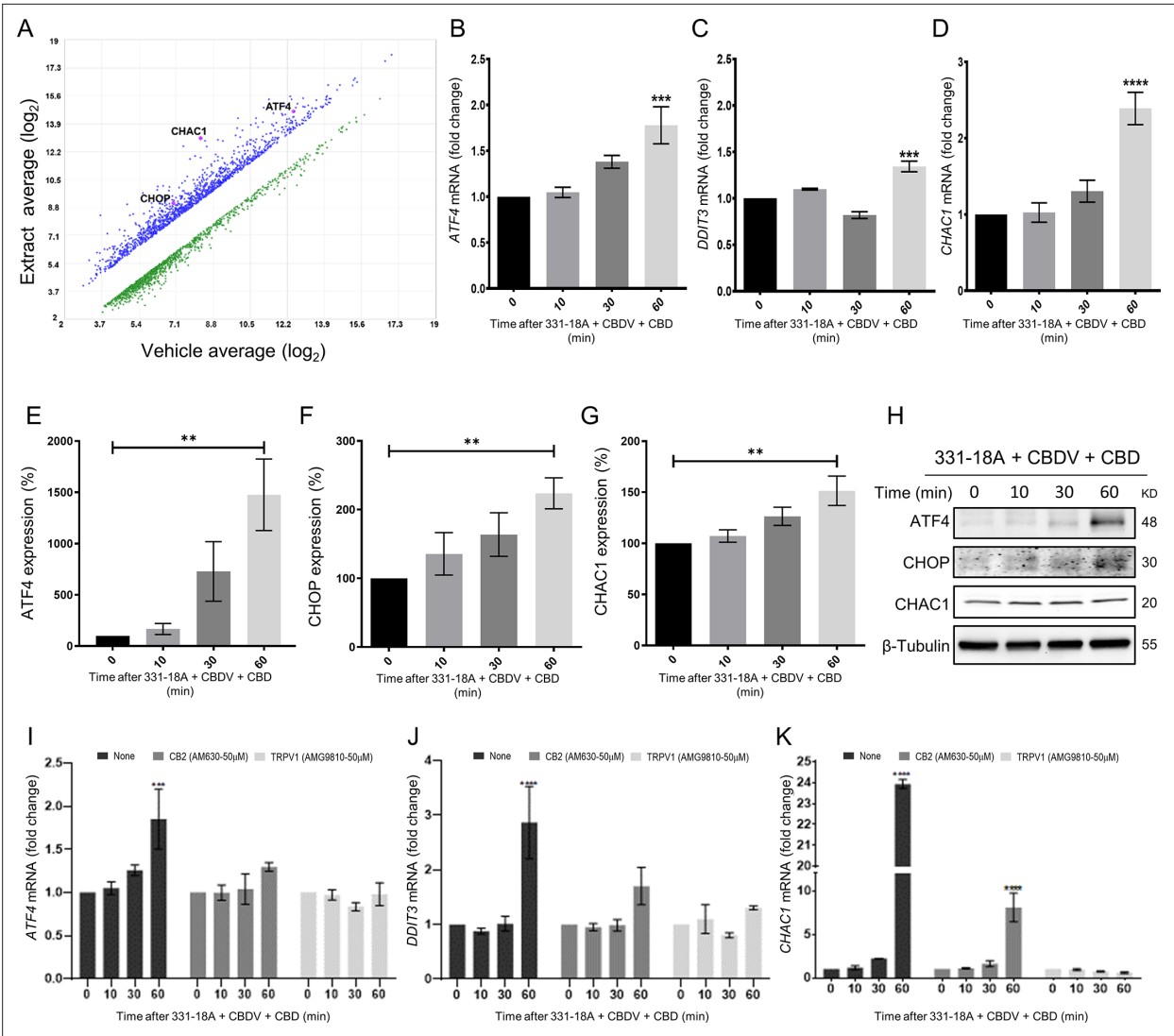

**Figure 3.** Notch1 downregulation by 331-18A, cannabidivarin (CBDV), and cannabidiol (CBD) combination is mediated via ATF4-CHOP-CHAC1 signaling pathway. (**A**) MOLT-4 cells were treated for 3 hr with either vehicle or the whole extract (3 μg/mL) and an Affymetrix scatter plot presents the differential expression; the increased-abundance genes *CHAC1*, *DDIT3*, and *ATF4* are marked (N=3). (**B–D**) MOLT-4 cells were treated for 3 hr with 331-18A, CBDV, and CBD combination (N=3), the mRNA expression of *ATF4*, *DDIT3*, and *CHAC1* genes was evaluated with qRT-PCR at different time points (0–60 min). Differences are presented as fold change ± SEM and statistically analyzed with one-way ANOVA (***p<0.001, ****p<0.0001). (**E–G**) MOLT-4 cells were treated as in B–D and the protein expression of ATF4, CHOP, and CHAC1 were evaluated with β-tubulin as the loading control. Intensity analysis was performed on three independent experiments (N=3) and statistically analyzed with an unpaired Student's t-test (**p<0.01). (**H**) Representative blots. (**I–K**) Analysis of *ATF4*, *DDIT3*, and *CHAC1* mRNA expression at different time points following pretreatment with CB2 and TRPV1 antagonists for 30 min and then treatment with the cannabinoid combination (N=3), analyzed with two-way ANOVA (***p<0.001, ****p<0.0001).

The online version of this article includes the following source data and figure supplement(s) for figure 3:

**Source data 1.** Original file for the western blot presented in *Figure 3H*.

**Figure supplement 1.** Notch1 downregulation by extract 12 is mediated through the ATF4-CHOP-CHAC1 signaling pathway.

**Figure supplement 1—source data 1.** Original file for the western blot presented in *Figure 3—figure supplement 1E*.

**Figure supplement 1—source data 2.** Original file for the western blot presented in *Figure 3—figure supplement 1I*.

*1I*, *Figure 3—figure supplement 1—source data 2*). These results demonstrate that CB2 and TRPV1 participate in inducing CHAC1 expression and preventing Notch1 maturation, resulting in a decrease in NICD release and cell death.

We therefore speculated that the combination of the three cannabinoids caused the activation of ATF4-CHOP-CHAC1 signaling by $Ca^{2+}$ depletion and ER stress (*Zhai et al., 2020*). To further verify

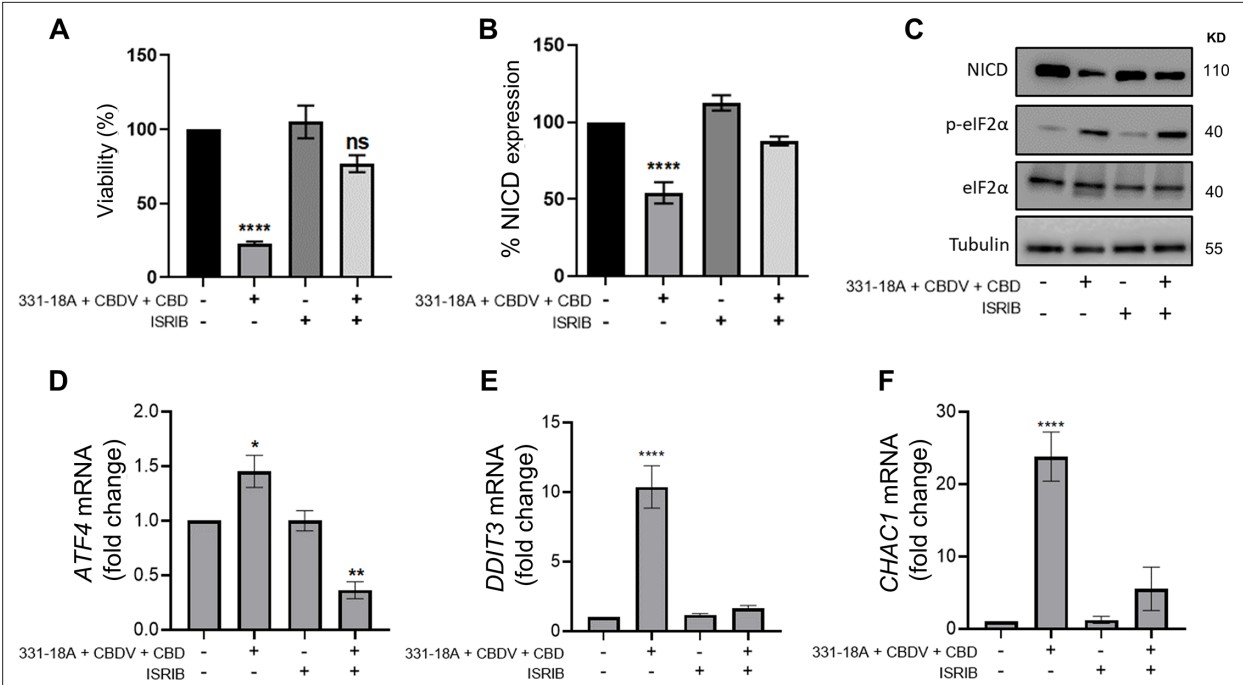

**Figure 4.** 331-18A, cannabidivarin (CBDV), and cannabidiol (CBD) combination activates eIF2α. MOLT-4 cells were pretreated for 30 min with the eIF2α inhibitor integrated stress response inhibitor (ISRIB) (150 μM) or left untreated, then treated with vehicle or the combination of 331-18A, CBDV, and CBD for 3 hr. (**A**) Viability after 24 hr was assessed with XTT (N=3). (**B**) Notch1 intracellular domain (NICD) protein expression was assessed (N=3) with β-tubulin as the loading control (unpaired Student's t-test, *p<0.05). (**C**) A representative image also showing the protein levels of phosphorylated eIF2α (Ser[51]) and total eIF2α. (**D–F**) *ATF4, DDIT3,* and *CHAC1* gene expression was assessed via qRT-PCR 4 hr after treatment (N=3). Results are presented as mean ± SEM and statistically analyzed with one-way ANOVA (****p<0.0001).

The online version of this article includes the following source data and figure supplement(s) for figure 4:

**Source data 1.** Original file for the western blot presented in *Figure 4C*.

**Figure supplement 1.** Whole extract treatment activates eIF2α.

**Figure supplement 1—source data 1.** Original file for the western blot presented in *Figure 4—figure supplement 1B*.

**Figure supplement 1—source data 2.** Original file for the western blot presented in *Figure 4—figure supplement 1E*.

the role of ATF4-CHOP-CHAC1 signaling in mediating the downregulation of the Notch1 pathway, we inhibited the activity of the upstream target eukaryotic initiation factor-2α (eIF2α) in MOLT-4 cells using integrated stress response inhibitor (ISRIB), which inhibits the activity of eIF2α without affecting its phosphorylation (***Halliday and Mallucci, 2015***; ***Figure 4—figure supplement 1A and B***, ***Figure 4—figure supplement 1—source data 1***). Then, we treated the cells with either vehicle or the combination of the cannabinoids. After 24 hr, blocking the activity of eIF2α prevented the cytotoxic effect of 331-18A, CBDV, and CBD combination (***Figure 4A***) or of the whole extract (***Figure 4—figure supplement 1C***). Moreover, eIF2α inhibition prevented the reduction in NICD expression upon treatment with either the whole extract (***Figure 4—figure supplement 1D and E***, ***Figure 4—figure supplement 1—source data 2***) or the cannabinoid combination (***Figure 4B and C***, ***Figure 4—source data 1***). Inhibition of eIF2α also rescued the cells from the induced increase in the mRNA expression of ATF4, DDIT3 (CHOP), and CHAC1 upon treatment with the combination of the cannabinoids (***Figure 4D–F***). Thus, inhibition of the integrated stress response pathway, eIF2α-ATF4-CHOP-CHAC1 signaling, rescued MOLT-4 cells from the cytotoxic effect of the combination of the three cannabinoids and prevented the reduction in NICD expression, as well as the increased transcription of ATF4, CHOP, and CHAC1 upon treatment.

To further investigate the interaction between 331-18A, CBDV, and CBD, we assessed their effect on the viability of MOLT-4 cells in different relative proportions. First we tested the ability of each separate cannabinoid to inhibit the proliferation of MOLT-4 cells in doses ranging 0–2 μg/mL (***Figure 5A***), which is below and above the overall concentration for which we found an effect for the cannabinoid

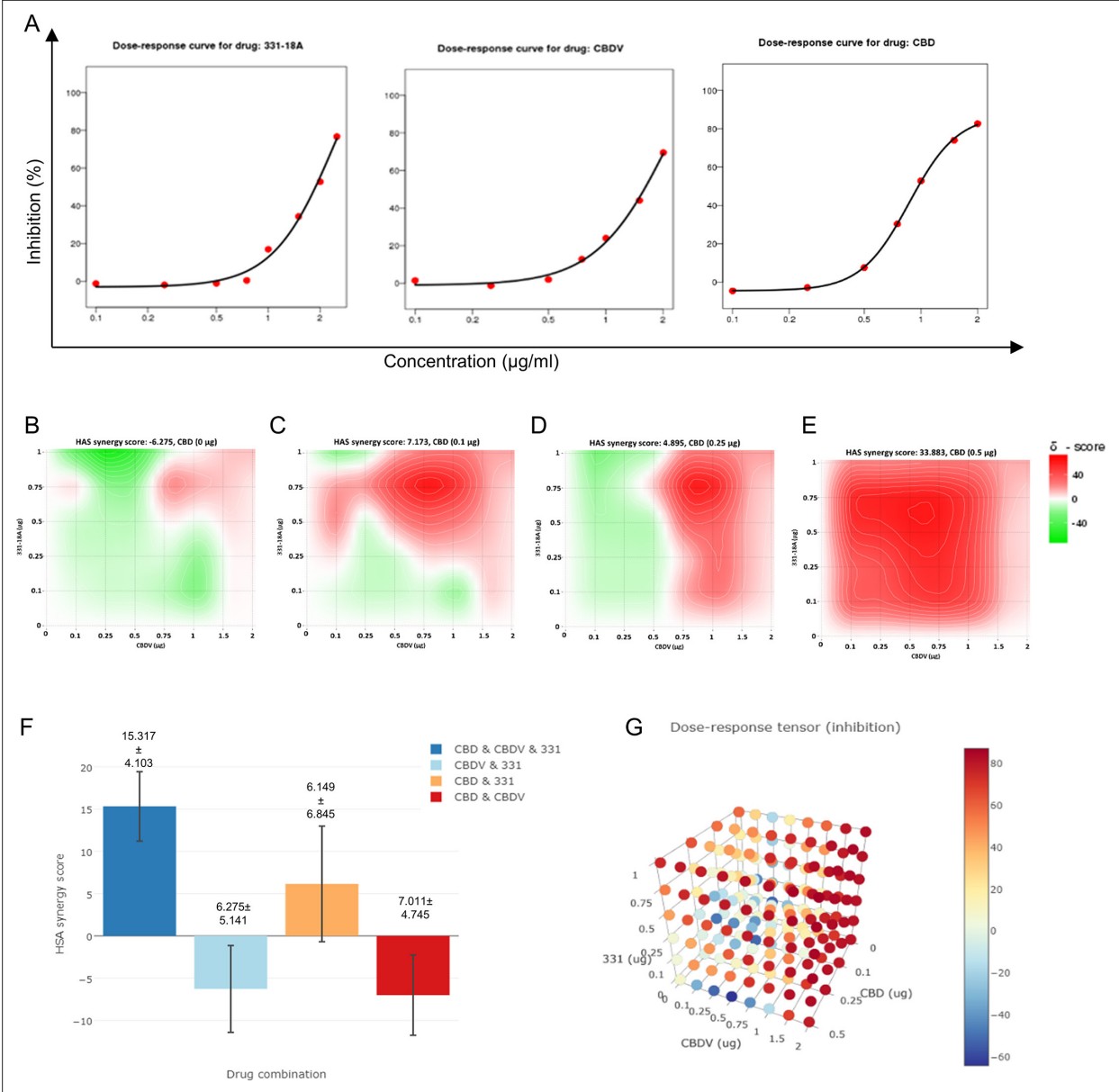

**Figure 5.** All three cannabinoids must be combined for a synergistic effect. (**A**) MOLT-4 cell death (N=3) was assessed by XTT following treatments with concentrations ranging 0–2 µg/mL and a dose-response curve for each cannabinoid separately was plotted. (**B–E**) Synergy distribution for different ratios of cannabinoids was calculated by Highest Single Agent (HSA) model with SynergyFinder web application (v 2.0), representative dose-response matrices are presented for cannabidiol (CBD) at (**B**) 0, (**C**) 0.1, (**D**) 0.25, and (**E**) 0.5 µg/mL. (**F, G**) Best synergy scores according to HSA model presented for the triple combination relative to the two-cannabinoid combinations.

combination. 331-18A and CBDV affected viability from a dose of 1 µg/mL and CBD from a dose of 0.5 µg/mL. Next, we calculated the combination index of every pair and all three cannabinoids in concentrations ranging from 0 to 2 µg/mL (*Figure 5B–E*). The most synergistic area is displayed in red. We calculated the total synergy score for the combination of all three cannabinoids relative to the pairwise combinations (*Figure 5F*). According to Highest Single Agent (HSA) reference model (*Yadav et al., 2015*), a combination of CBDV with either 331-18A or CBD results in a likely antagonistic interaction (score < –10), a combination of 331-18A and CBD results in a likely additive interaction (score from –10 to 10), and only the combination of all three cannabinoids is likely synergistic (score >10, *Figure 5G*).

## Inhibition of cancer progression in vivo by 331-18A, CBDV, and CBD combination

We further evaluated the antitumor properties of the three cannabinoids using several mouse models. The first model was a subcutaneous tumor model for which we have already demonstrated the effectivity of the whole extract (*Besser et al., 2023*). We engrafted subcutaneously nonobese diabetic-severe combined immunodeficiency (NOD/Scid) mice with MOLT-4 cells, then treated with vehicle only or the combination of the cannabinoids from when tumors reached a volume of 100 mm$^3$ (day 7), on alternating days, for a duration of 4 weeks. Treatment with a combination of 331-18A, CBDV, and CBD significantly inhibited tumor growth (*Figure 6A–C*). Significant differences were apparent approximately 3 weeks after treatment started (*Figure 6A*), and at the endpoint, 34 days from the injection of cells, the average weight of tumors was significantly lower in the treated group compared to the control group (*Figure 6B*). Excised tumors (*Figure 6C*) were analyzed for their cannabinoid content according to a method recently developed by our group for biological tissues (*Berman et al., 2020*). Using ESI-LC/MS, we identified and quantified each of the three cannabinoids in the tumors (16.0±6.7, 22.3±10.4, and 265.8±109.7 ng/g for CBDV, 331-18A, and CBD, respectively [average ± SEM, N=3]). Notably, the ratio of the cannabinoids analyzed in the tumors was similar to the ratio given to the mice in the treatment protocol. In addition, a significant decrease was observed in NICD expression in the tumors treated with the cannabinoids (*Figure 6D*, *Figure 6—source data 1*). To assess possible toxicity, the body weight of the mice was monitored throughout the experiment (*Figure 6E*). There were no significant differences in the average weight of mice in the control group and the treatment group.

Next, we assessed the cannabinoids using an intravenous model which better resembles leukemogenesis (*Lee et al., 2012*; *Xin et al., 2022*). We engrafted NOD/Scid IL2rγ-null (NSG) mice that are a better host for human immune cells with CCRF-CEM, Notch1-mutated T-ALL cells. The mice were treated with a combination of the three cannabinoids, and to verify the need for the combination, one group was also treated with pure CBD in the same overall dose. At the endpoint of the experiment, the amount of human CD45+ cells detected in the bone marrow (BM) was significantly decreased in the mice treated with the combination of the three cannabinoids compared to the control, and compared to pure CBD (*Figure 6F*). In this model as well, there were no significant differences in the average weight of the mice (*Figure 6G*). The combination of 331-18A, CBDV, and CBD was as effective as the whole extract (*Figure 6—figure supplement 1A and B*). In addition, the administration of the cannabinoid combination significantly prolonged survival in treated mice compared to the survival of vehicle-treated control mice (*Figure 6H*), to a similar degree to the whole extract (*Figure 6—figure supplement 1C*).

To examine the clinical relevance, we utilized a patient-derived xenografts (PDX) model with primary cells from a T-ALL patient. After allowing 35 days from transplantation for leukemic cell establishment, mice were treated for 3 weeks with either vehicle or whole extract containing the three cannabinoids. In mice that were treated with the extract, we found a significant reduction in the burden of leukemia in the BM (*Figure 6—figure supplement 2A*), peripheral blood (*Figure 6—figure supplement 2B*), and spleen (*Figure 6—figure supplement 2C*). The weight of the spleen was also significantly lower in extract-treated mice (*Figure 6—figure supplement 2D and E*). Treatment with the whole extract did not affect the weight of the mice (*Figure 6—figure supplement 2F*).

## Discussion

Targeting Notch signaling components has generated much interest for its therapeutic potential. However, so far efforts to develop treatments that target Notch signaling have been unsuccessful (*Previs et al., 2015*). Cannabis, in its many forms, is already being prescribed to cancer patients all over the world, mainly as a palliative treatment for the inhibition of nausea and emesis associated with chemotherapy, appetite stimulation, pain relief, and relief from insomnia (*Abrams and Guzman, 2015*). On top of that, recent studies have demonstrated that phytocannabinoids also possess anticancer potential (*McAllister et al., 2015*; *Hinz and Ramer, 2022*; *Baram et al., 2019*; *Russo and Marcu, 2017*; *Velasco et al., 2016*; *Guzmán, 2003*).

In the current study, we identified three unique phytocannabinoids that together selectively induce apoptosis in Notch1-mutated leukemia cells: CBD, CBDV, and the novel phytocannabinoid 331-18A.

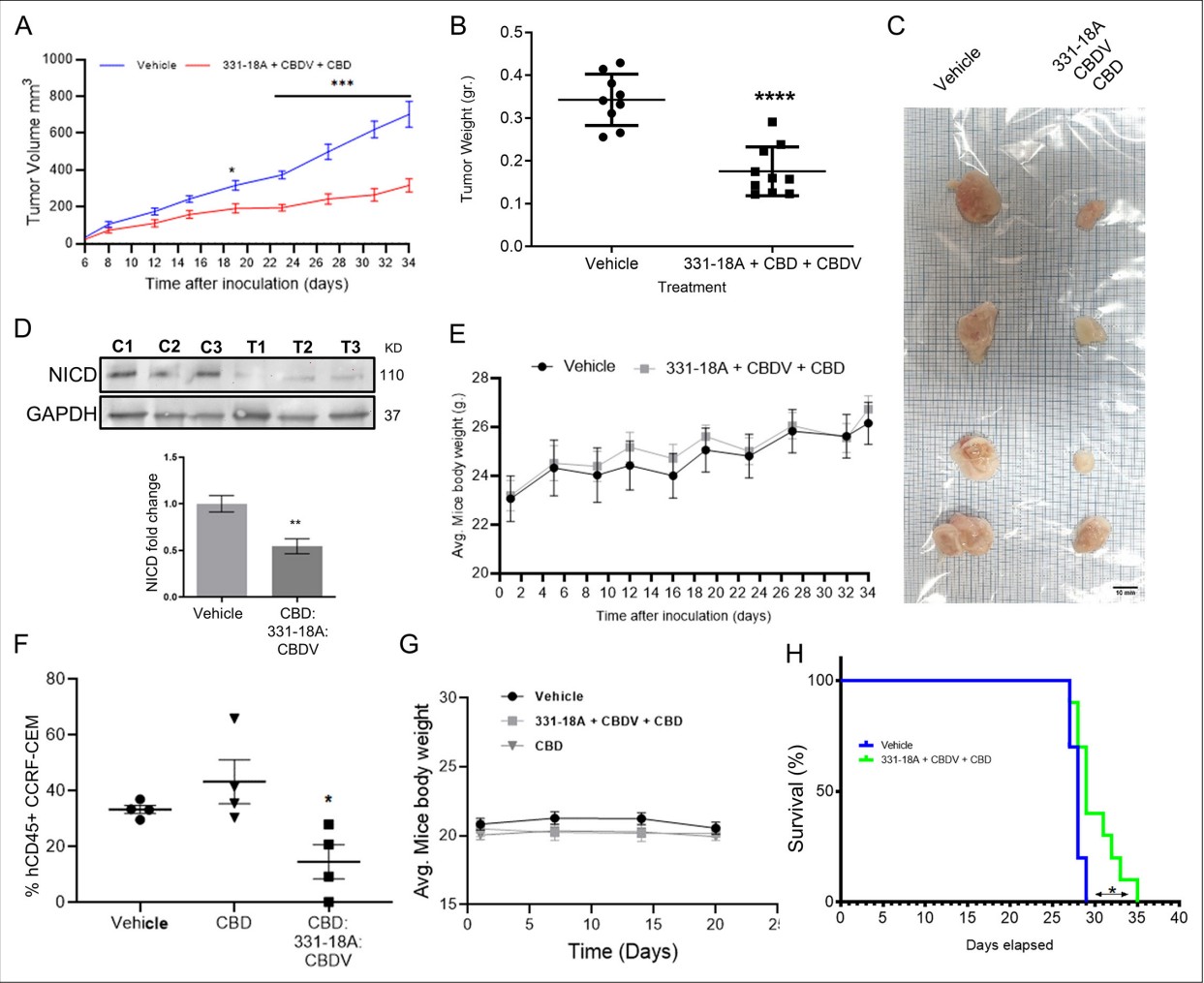

**Figure 6.** Inhibition of cancer progression in vivo by a combination of 331-18A, cannabidivarin (CBDV), and cannabidiol (CBD). (**A**) Female nonobese diabetic-severe combined immunodeficiency (NOD/Scid) mice (N=4–5/group, two independent experiments) were engrafted subcutaneously with $1\times10^6$ MOLT-4 cells. After 7 days, the mice were randomly divided into two groups and alternate-day treated intraperitoneally with either vehicle or a combination of 331-18A (1.11 mg/kg), CBDV (1.11 mg/kg), and CBD (17.77 mg/kg). Ectopic tumor volume was measured using a vernier caliper and calculated according to the formula (length × width$^2$)×0.5. The average difference between groups was statistically analyzed by Bonferroni's multiple comparisons test (*p<0.05, ***p<0.001). (**B**) After 34 days of treatment, the mice were sacrificed and tumors were excised and weighed (N=9). The difference in the weights of the tumors between treatment groups was statistically analyzed by unpaired Student's t-test (****p<0.0001). (**C**) Representative photograph of excised tumors. (**D**) A representative blot (*top*) and fold change (*bottom*) of Notch1 intracellular domain (NICD) protein expression in excised tumors at day 34 after treatment with either vehicle (C) or a combination of the three cannabinoids (T), with GAPDH as the loading control. Data are presented as mean ± SEM (N=3) and statistically analyzed by unpaired Student's t-test (**p<0.01). (**E**) Body weight (grams) in vehicle- and cannabinoids-treated mice. (**F**) Female NSG mice were intravenously injected with $1\times10^6$ human CCRF-CEM cells. Mice were randomly divided into three groups (N=5/group) and treated intraperitoneally daily for 5 days, then left untreated for 2 days before starting another cycle, for a period of 3 weeks. Treatment groups were vehicle, pure CBD (30 mg/kg), and a combination of CBD (26.67 mg/kg), CBDV (1.67 mg/kg), and 331-18A (1.67 mg/kg) at the same overall concentration as pure CBD. After 24 days, the percentage of hCD45+ cells in the bone marrow was measured with flow cytometry and statistically analyzed by unpaired Student's t-test (*p<0.05). (**G**) Body weight (grams) in vehicle- and cannabinoids-treated mice. (**H**) Survival analyses after CCRF-CEM injection followed by treatment with either vehicle or the cannabinoid combination (N=10/group), statistical differences were calculated with the Log-rank (Mantel-Cox) test (*p<0.05).

The online version of this article includes the following source data and figure supplement(s) for figure 6:

**Source data 1.** Original file for the western blot presented in *Figure 6D*.

**Figure supplement 1.** Inhibition of tumor growth in vivo by whole extract.

**Figure supplement 2.** Treatment with extract containing the three cannabinoids inhibits leukemic expansion in an in vivo patient-derived xenografts (PDX) model.

CBD and CBDV are well known and commercially available. However, to the best of our knowledge, the third phytocannabinoid 331-18A, denoted in the literature as 6,12-dihydro-6-hydroxy-cannabid iol (*Stoss and Merrath, 1991*), has not been previously identified in natural and/or decarboxylated Cannabis plants. We suggest that the need for all three cannabinoids for the optimal antitumor effect demonstrates the critical importance of elucidating the individual and polypharmacological effects of the entire molecular spectrum of Cannabis.

Although CBD by itself demonstrated some capacity to induce cell death, it was only when all three cannabinoids were combined that the same extent of response was achieved as treating with the whole extract. The Cannabis plant produces a vast array of over 140 phytocannabinoids and hundreds of other components, but the full extent of their biological effects and interactions is still not completely understood. Nevertheless, mounting evidence suggests that their combined effect as a whole is greater than that of each separate compound (*Russo, 2011*; *Worth, 2019*). Recently, *Blasco-Benito et al., 2018*, have shown that a $\Delta^9$-THC-rich extract was much more potent than pure $\Delta^9$-THC in producing antitumor responses in both cell culture and animal models of breast cancer. However, they were unable to identify the combinations of compounds that were responsible for this increased potency. In previous publications by our group, we showed that different CBD-rich Cannabis extracts, with equal amounts of CBD but varying concentrations of other minor compounds, led to diverse anticonvulsant effects in a mouse model of epilepsy (*Berman et al., 2018*), and selective antitumoral effects in Notch1-mutated in T-ALL (*Besser et al., 2023*), suggesting the potential therapeutic effects of other compounds in the extracts in addition to CBD.

A possible mechanism previously suggested to explain the difference between the effects of purified phytocannabinoids versus full-spectrum extracts is the 'entourage effect', where one compound may synergize the activity and efficacy of another on the same target (*Russo, 2011*; *Worth, 2019*). This 'entourage effect' was postulated by *Russo, 2011*, based on a phenomenon observed in the eCBS by *Ben-Shabat et al., 1998*. While it is well established for endocannabinoids, only very few studies demonstrated this for phytocannabinoids. We showed here that the presence of small amounts of the minor cannabinoids 331-18A and CBDV enhanced the effect of CBD by itself. Our success in isolating and characterizing a minimal number of cannabinoids that achieve the same antitumor effect as the whole extract is a significant advancement in the field of medical Cannabis and provides solid evidence for the 'entourage effect'.

Subsequent reports have expanded the notion of the 'entourage effect' to polypharmaceutical and/or polypharmacological effects, where several of the plant metabolites work simultaneously on the same receptors, and/or when the same molecule has several different targets within the same system (*Mechoulam and Ben-Shabat, 1999*; *Russo, 2018*; *Di Marzo and Piscitelli, 2015*). CBD has been shown to present complex signaling capabilities of many receptors in the extended eCBS. Specifically, it was shown to have a weak affinity to CB2, and a strong affinity to many of the TRPV family receptors, including TRPV1, TRPV2, and TRPV3 (*Kis et al., 2019*; *De Petrocellis et al., 2012*).

We suggest that the Notch1-mutated T-ALL cells are affected through the engagement of the eCBS receptor CB2 and the channel TRPV1. More specifically, the known cannabinoids CBD and CBDV affected both, while the novel 331-18A exerted its effect only through CB2. Following the stimulation of the receptors, $Ca^{2+}$ is depleted from the ER and there is an increase in cytosolic $Ca^{2+}$ and activation of eIF2α followed by ATF4-CHOP-CHAC1 signaling. Notably, the inhibition of eIF2α upstream to ATF4 rescued T-ALL cells from reduced NICD expression by treatment with 331-18A, CBDV, and CBD combination. The upregulation of CHAC1 inhibits the S1 furin-like cleavage of Notch1 at the Golgi apparatus (*Chi et al., 2012*), leading to the arrival of immature Notch1 to the plasma membrane as we have previously shown (*Besser et al., 2023*). This inhibition of the Notch1 pathway ultimately leads to apoptosis in T-ALL cells that rely on mutated Notch1 for propagation (see schematic representation in *Figure 7*).

The potential antitumor effects of cannabinoids through the Notch1 signaling pathway were previously demonstrated in glioma, where THC induced the expression of p8 protein, upstream to ATF3/4 (*Carracedo et al., 2006*). The activation of Notch signaling by ER stress was demonstrated in additional systems, including Alzheimer's disease (*Ohta et al., 2011*), polycystic ovary syndrome (*Koike et al., 2023*), and other types of cancer (*Izrailit et al., 2017*; *Ding et al., 2018*). A previous work that focused on chronic lymphocytic leukemia suggested a role for ER stress in Notch1 cleavage and leukemia cell death through the inhibition of proteasome activity (*Rosati et al., 2013*). These findings

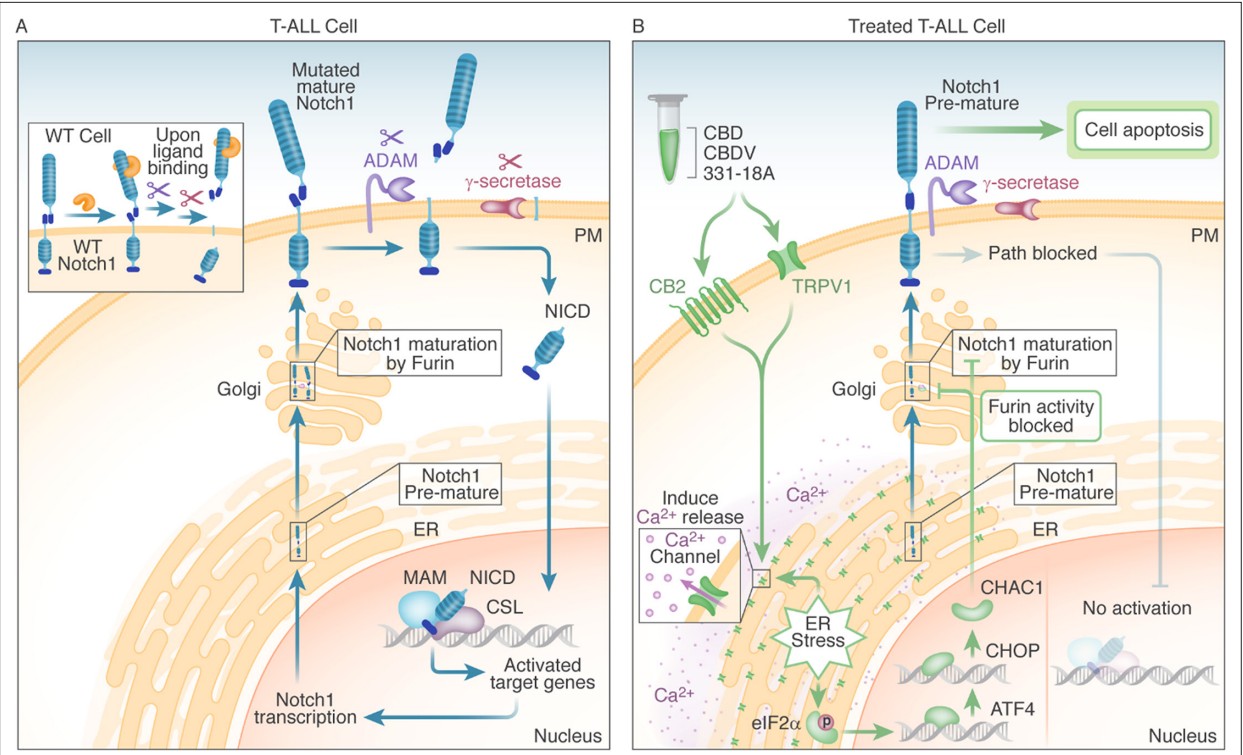

**Figure 7.** Schematic representation. (**A**) In WT cells, the Notch1 receptor is only activated upon ligand binding. In T-cell acute lymphoblastic leukemia (T-ALL) cells that have a Notch1 mutation, the receptor is constitutively active regardless of ligand binding, leading to a positive feedback loop in which the mature receptor is cleaved by a series of sequential cleavage events, causing the release of Notch1 intracellular domain (NICD), which translocates to the nucleus where it promotes transcription of target genes involved in cell growth. The premature Notch1 is cleaved by a furin-like convertase in the trans-Golgi apparatus, resulting in a mature receptor that is transported to the membrane. (**B**) The three phytocannabinoids cannabidiol (CBD), cannabidivarin (CBDV), and 331-18A in the whole extract stimulate CB2 and TRPV1 leading to an increase in cytosolic $Ca^{2+}$ and ER stress-induced activation of $eIF2\alpha$ followed by ATF4-CHOP-CHAC1 signaling. CHAC1 inhibits the furin-like cleavage of Notch1, preventing its maturation and resulting in reduced NICD formation, reduced cell viability, and increased apoptosis.

suggest blocking Notch1 maturation through the $eIF2\alpha$-ATF4-CHOP-CHAC1 pathway and inhibiting the S1 furin-like cleavage of Notch1 have the potential to affect all types of activation mutations in Notch1, and also all four human homologs of Notch (Notch1-4) (*Chi et al., 2012*; *Mungrue et al., 2009*).

We demonstrated the effectiveness of the whole extract and the combination of the cannabinoids in several in vivo models. To validate the clinical relevance of our findings for patients, we tested the extract containing the three cannabinoids in an in vivo PDX model (*Jung et al., 2018*; *Richter-Pechańska et al., 2018*; *Figure 6—figure supplement 2*), and demonstrated treatment reduced leukemic burden. As medical Cannabis is already routinely prescribed to cancer patients, tailoring chemovars that contain the appropriate cannabinoids to T-ALL patients can provide them with optimal care that offers anti-cancer properties on top of the palliative ones. Our findings are potentially relevant for numerous diseases, as dysregulation of Notch signaling has been found in various other cancers including breast, prostate, and colorectal, as well as in non-cancerous diseases (*Qiu et al., 2018*; *Rice et al., 2019*; *Vinson et al., 2016*). As such, they may pave the way for the establishment of new pharmacotherapies for the treatment of Notch1-dependent malignancies and other diseases.

## Materials and methods
### Reagents
The analytical standards (>98%) for phytocannabinoids were purchased from Silicol Scientific Equipment Ltd. (Or Yehuda, Israel). The antagonists CID 16020046 (#4959/5), AM630 (#1120/10), SR-144,528 (#5039/10), and AMG9810 (#2316/10) were purchased from Tocris Bioscience (Bristol, UK). BIM-46187

(#5332990001) and ISRIB (#SML0843) were purchased from Sigma-Aldrich (Rehovot, Israel). Synthetic CBD, CBDV, and 331-18A were purchased from Cannasoul Analytics (Caesarea, Israel).

## Phytocannabinoids analysis

Comprehensive phytocannabinoid analysis was performed using a UHPLC system coupled to a Q Exactive Focus Hybrid Quadrupole-Orbitrap MS (Thermo Scientific, Bremen, Germany) using a similar chromatographic method as previously described (*Berman et al., 2018*; *Besser et al., 2023*). MS acquisition was carried out with a heated electrospray ionization (HESI-II) ion source operated in negative mode. Source parameters were as follows: sheath gas flow rate, auxiliary gas flow rate, and sweep gas flow rate: 50, 20, and 0 arbitrary units, respectively; capillary temperature: 350°C; heater temperature: 50°C; spray voltage: 3.00 kV. The scan range was 150–550 m/z for all acquisition events. MS was operated in full MS$^1$ mode at 70,000 resolution, and the AGC target was set to $10^6$ with a maximum IT of 100 ms. Identification and absolute quantification of phytocannabinoids was performed by external calibrations as described by *Berman et al., 2018*. Absolute quantification of phytocannabinoids with analytical standards was performed by external calibrations. Standard mixes were prepared ranging from 1 to 1000 ng/mL for $\Delta^9$-THCA, $\Delta^9$-THC, CBDA, and CBD, and 0.25–625 ng/mL for all the other components. The dynamic range for each component was determined as maximum deviation from expected concentrations of 20%, and minimum signal-to-noise ratios of 10. All other phytocannabinoids without analytical standards were semi-quantified according to calibration curves from the same phytocannabinoid family or average curves for unknown compounds (*Berman et al., 2018*). All samples were injected and analyzed by ESI-LC/MS in three concentrations (10, 1, and 0.1 µg/mL Cannabis extract to ethanol).

## Isolation of single phytocannabinoids

Fractionation of the whole Cannabis extract into four fractions and isolation of single phytocannabinoids were performed using a semi-preparative HPLC/UV (Thermo Scientific, Bremen, Germany). The chromatographic separation was achieved using a Luna C18 column (10 µm, 250 mm × 21.1 mm i.d.) and a two-solvent A/B multistep gradient (solvent A: 0.1% acetic acid in water and solvent B: 0.1% acetic acid in acetonitrile, the solvents and acetic acid were of HPLC and LC/MS grades, respectively). The multistep gradient program was established as follows: initial conditions were 50% B for 4 min, raised to 76% B until 6.5 min, held at 76% B for 16.5 min, and then raised to 90% B until 26 min, held at 90% B until 31 min, decreased to 50% B over the next 4 min, and held at 50% B until 40 min for re-equilibration of the system prior to the next injection. A flow rate of 20 mL/min and an injection volume of 500–750 µL were used. Data acquisition was performed at 220 nm. The crude Cannabis extract was prepared in a concentration of 100 mg/mL in ethanol. Fractions were collected in 10 min intervals and then lyophilized to dryness and analyzed by UHPLC/UV and ESI-LC/MS.

To improve the yield and purity of the fractions for isolation of single phytocannabinoids, a preliminary separation step was employed prior to semi-preparative HPLC fractionation. The crude Cannabis extract was dissolved to a concentration of 0.5 g/mL in ethanol, filtered using a PTFE 0.45 µm filter, and then purified using a PLC 2050 Purification System, equipped with a CPC-250 column and a photodiode array detector (Gilson Inc). Separations were achieved using a biphasic solvent system consisting of upper (hexane:acetonitrile in the ratio of 98:2 vol/vol) and mobile (acetonitrile:water in the ratio 50:50 vol/vol) phases in descending mode. The column was operated with a flow rate of 12.5 mL/min and a rotation speed of 2200 rpm. The gradient program was established as follows: initial conditions were acetonitrile:water:hexane in the ratio 50:50:0 vol/vol for 25 min, and then gradually raised to acetonitrile:water:hexane in the ratio 64:34:2 vol/vol until 45 min. All the solvents were of HPLC grade. The collected fractions were lyophilized to dryness and then subjected to semi-preparative HPLC fractionation. The specific phytocannabinoids were collected according to retention time.

## Structure elucidation

The structure of the isolated 331-18A was elucidated by $^1$H and $^{13}$C one-dimensional NMR spectra (Avance-III-700, Bruker, Germany) with CBD as the reference. The samples were dissolved in CDCl$_3$ and analyzed at 4°C with tetramethylsilane as the internal reference. The optical rotation of 331-18A

in comparison to CBD was determined in chloroform (10 mg/mL) using a Jasco P-1010 (Rudolph Research Analytical, USA) polarimeter at 26°C.

## Cell culture

The T-cell leukemia cell lines MOLT-4 (RRID:CVCL_0013, CRL-1582) and CCRF-CEM (RRID:CVCL_0207, CCL-119) were purchased from ATCC, where their identity has been authenticated by COI assay and STR analysis. Cell lines were grown in RPMI 1640 media (Sigma-Aldrich; R8758), and kept at 37°C in a 5% carbon dioxide incubator. Unless otherwise stated, all culture media were supplemented with 10% FBS (Biological Industries; 04-007-1A), penicillin (100 U/mL), and streptomycin (100 µg/mL) (Biological Industries; 03-031-1B). Cell lines were confirmed as negative for mycoplasma contamination using EZ-PCR Mycoplasma Detection Kit (Biological Industries, USA).

## Cell apoptosis assay

Cells were cultured in 12-well plates at $1 \times 10^6$ cells/well with media containing 0.5% FBS and incubated for 24 hr with either DMSO (vehicle), Cannabis extract 12, fractions 1–4, and different combinations of 0.06 µg/mL 331-18A, 0.06 µg/mL CBDV, and 1.5 µg/mL CBD. Apoptotic cells were detected by an Annexin V/PI assay using flow cytometry or by detection of cleaved caspase-3.

## Annexin V/PI assay

Apoptosis was assessed by Annexin V-FITC (BioVision; 1006-200, 1:500) and PI staining in an annexin binding buffer (BioVision; 1006-100; 1:500) according to the manufacturer's instructions. Apoptosis assessed by flow cytometry used a BD LSR II digital four-laser flow cytometer (BD Biosciences) and was analyzed by BD FACSDiva software version 6.1.2. (BD Biosciences). Results were calculated as percentage of positive Annexin V-FITC cells out of total cells counted (30,000 events).

## Western blot analysis

Cells were quickly washed twice with ice-cold PBS, and lysed in radioimmunoprecipitation assay lysis buffer (Sigma-Aldrich; R0278) freshly supplemented with protease and phosphate inhibitors (Cell Signaling Technology; 5872). Lysates were cleared by centrifugation at 13,000 × $g$ at 4°C for 10 min. Protein concentration was determined by a Bradford assay (Sigma-Aldrich; B6916). Proteins were denatured in NuPAGE LDS Sample Buffer (4×) (Thermo Fisher; NP0008) and supplemented with β-mercaptoethanol (Sigma-Aldrich; B6131). Typically, 20–50 mg of total protein were loaded per lane on a Novex 4–20% Tris-Glycine gradient gel (Invitrogen; XP04202BOX) and electrophoretically transferred to a nitrocellulose membrane (Bio-Rad; 1704159S). Membranes were blocked with TBS 0.1% Tween 20 buffer containing 5% bovine serum albumin (Sigma; A7906) for 1 hr and incubated with the primary antibody overnight at 4°C. This was followed by incubation with horseradish peroxidase-labeled matching secondary antibodies. Immunoreactive bands were detected by Luminata HRP substrate (Millipore; WBLUR0500) and visualized using a MicroChemi imager (DNR Bioimaging Systems). Antibodies used: Cleaved Notch1 Val1744 (CST, 4147S; RRID:AB_2153348; 1:1000) to assess NICD, cleaved caspase-3 (CST, 9664; RRID:AB_2070042; 1:1000), β-tubulin (CST; clone D3U1W, 86298; RRID:AB_2715541; 1:1000), GAPDH (CST, 2118; RRID:AB_561053; 1:5000), ATF4 (Proteintech, 10835-1-AP; RRID:AB_2058600; 1:500), CHOP (Proteintech, 15204-1-AP; RRID:AB_2292610; 1:500), CHAC1 (abcam, ab76386; RRID:AB_1658635; 1:500), eIF2α (CST, 5324; RRID:AB_10692650; 1:1000) and Phospho-eIF2α (Ser51) (CST, 3398; RRID:AB_2096481AB_2096481; 1:1000). Secondary antibodies include: Peroxidase AffiniPure Goat Anti-Rabbit IgG (H+L) (Jackson ImmunoResearch, 111-035-144; RRID:AB_2307391; 1:5,000) and Peroxidase-AffiniPure Goat anti mouse IgG (HZ +L) (Jackson ImmunoResearch, 115-035-003; RRID:AB_10015289; 1:5000).

## RNA extraction

Total RNA was isolated from MOLT-4 cells ($1 \times 10^6$ cells/sample) using TRIzol (Thermo Fisher; 15596026) and RNeasy kit (QIAGEN; 74104) according to the manufacturer's instructions. Sample quality was assessed by both spectrophotometrically (NanoDrop Technologies) and gel electrophoresis with 1% agarose.

## Affymetrix microarray

RNA sequencing of differentially expressed genes was performed on extracted RNA with an Affymetrix Clariom-S microarray for human genome RNA expression analysis and an Affymetrix Transcriptome Analysis Console software. The threshold set for increased- and decreased-abundance genes was a fold change $\geq 1.5$ and $p < 0.05$. The sequencing was performed by the Center for Genomic Technologies of The Hebrew University of Jerusalem.

## Quantitative RT-PCR

cDNA was synthesized from 1 µg of extracted RNA. Purified RNA was reverse-transcribed with the qScript cDNA synthesis kit (Quanta Biosciences; 95047). The mRNA expression levels were quantified using qRT-PCR 7300 system (Applied Biosystems) using Human Taqman probes (Thermo Fisher Scientific) for: *GAPDH* (Hs02758991_g1); *CNR1* (CB1) (Hs01038522_s1); *CNR2* (CB2) (Hs00361490_m1); *GPR55* (Hs00271662_s1); *TRPV1* (Hs00175798_m1); *TRPM8* (Hs01066596_m1); *TRPA1* (Hs00175798_m1); *ATF4* (Hs00909569_g1); *DDIT3* (CHOP) (Hs00358796_g1); *CHAC1* (Hs00225520_m1). Relative expression values were normalized using the endogenous housekeeping gene *GAPDH* as the control and calculated using standard ΔCT methods.

## Calcium imaging

MOLT-4 cells were loaded with the fluorescein $Ca^{2+}$ probe Fluo-4 AM (Invitrogen #F14217). Cells were incubated for 60 min in RPMI medium and supplemented with 1 µM Fluo-4 AM. Cells were then rinsed twice and incubated for another 30 min in a Fluo-4 AM-free Hanks' Balanced Salt Solution (HBSS) buffer (Sigma-Aldrich #H6648). During these steps the cells were kept in the dark. Fluo-4 signals were collected by a BD LSR II digital four-laser flow cytometer (BD Biosciences) and analyzed by BD FACS-Diva software version 6.1.2 (BD Biosciences).

## Cell viability assay

Cells were cultured in 96-well plates. At $1 \times 10^5$ cells/well the media was replaced with RPMI containing 0.5% FBS and treatments as indicated. DMSO was used as the vehicle. The XTT-based cell proliferation kit (Sartorius, Beit Haemek, Israel) was used to quantitate viability according to the manufacturer's instructions. Cytotoxicity was measured following 24 hr of incubation using fluorescence spectrophotometry at excitation and emission wavelengths of 450–500 and 630–690 nm, respectively, and calculated as percent reduction between treated and control cells.

## Synergy scoring

The cell viability percentage for each concentration was assessed with an XTT-based cell proliferation kit (Sartorius, Beit Haemek, Israel), then entered into SynergyFinder web application (v 2.0). The HSA model was used to calculate synergy scores of the cannabinoid combinations. Estimation of outlier measurements was carried out with cNMF algorithm (*Ianevski et al., 2019*) implemented in SynergyFinder.

## Mice and procedures

All procedures were performed according to the protocols approved by the Technion Administrative Panel of Laboratory Animal Care (animal protocol number: IL_0470317). Female NOD.CB17-Prkdcscid/NCrHsd (IMSR_ENV:HSD-170) adult (6–7 weeks) mice were purchased from Envigo. Female NSG adult (6–7 weeks) mice were purchased from the Technion animal facility.

For the in vivo tumorigenicity assay, MOLT-4 cells mixed with BD Matrigel Matrix Growth Factor Reduced (BD; FAL356230) (1:1 vol/vol) were injected subcutaneously into the flanks of 7-week-old female NOD/Scid mice. Tumor size was measured with a caliper, and the tumor volume was determined according to the formula: length × width$^2$ × 0.5. The mice were euthanized after the indicated days as none reached the maximal endpoint size (1 cm$^3$). The xenograft tumors were dissected and weighed.

For the in vivo leukemogenesis assay and survival analysis, CCRF-CEM cells were intravenously injected to 7-week female NSG mice. The BM was harvested from each mouse and the infiltrated CCRF cells were identified by staining for a human CD45 antibody (BioLegend; RRID:AB_2687375; BLG-368522; 1:200).

For the PDX model, frozen aliquots of T-ALL male patient samples in which Notch1 mutation was verified by next-generation sequencing were purchased from Dana-Farber Cancer Institute. The cells were expanded in mice and injected subcutaneously into the flanks of 7-week female NSG mice. To investigate the inhibitory effects of extract 12 on leukemogenesis, 30 mg/kg/day extract was intraperitoneally injected three times a week, starting 3 weeks after cell injection. The mice were euthanized after the indicated days. BM, peripheral blood, and spleens were harvested from each mouse and the infiltrated primary cells were identified using flow cytometry by staining for human CD45 antibody (BioLegend; BLG-368522; 1:200).

## Cannabinoids analysis from tumor tissue

Fresh excised tumors were accurately weighed and dissected to smaller sections. HBSS (Sigma-Aldrich; H8264) was added in a ratio of 1 ml per 100 mg sample. The samples were further homogenized into single-cell suspensions with a gentleMACS Dissociator (Miltenyi Biotec) using a pre-set program for tumor tissues. The single-cell suspensions were stored at –80°C until further extraction. CBD, CBDV, and 331-18A were analyzed using the ESI-LC/MS system by retention time, accurate mass, and spectral matching against our developed ESI-LC/MS/MS library of phytocannabinoids (*Berman et al., 2018*). Absolute quantification of the three compounds was performed by the stable isotope dilution method as recently described by our group (*Berman et al., 2020*). Briefly, the extraction solution (methanol:acetonitrile:acetic acid in the ratio 50:50:0.1 vol/vol) spiked with 20 ng/mL of CBD-d3 internal standard (Sigma-Aldrich; C-084) was added to the single-cell suspensions in the ratio 3:1 vol/vol, respectively. Samples were thoroughly vortexed and then centrifuged for 20 min at 4°C for protein and cell precipitation. The supernatants were then mixed in a ratio of 1:3 vol/vol sample to 0.1% vol/vol acetic acid in water, and loaded onto Agela Cleanert C8 solid phase extraction cartridges (500 mg of sorbent, 50 µm particle size). Cannabinoids were eluted from the columns using 2 mL of 0.1% vol/vol acetic acid in methanol, evaporated to dryness by SpeedVac, and reconstituted in 100 µL ethanol. Since there was no analytical standard available for 331-18A, this compound was semi-quantified according to the calibration curve of CBD.

## Statistics and reproducibility

Data were analyzed with GraphPad Prism software version 7.04. Statistical significance was determined by either an unpaired two-tailed Student's t-test (with unequal variations if an F test ruled out the equal variation assumption) or one-way ANOVA followed by Sidak's multiple comparisons test for multiple groups. Kaplan-Meier survival curves were analyzed with the Gehan-Breslow-Wilcoxon test, as appropriate for the dataset. Data are generally expressed as mean ± SEM, as denoted in the figure captions, with $p \leq 0.05$ was considered statistically significant.

## Acknowledgements

We thank the the Israel Cancer Research Fund, Evelyn Gruss Lipper Charitable Foundation, and the Israeli Ministry of Agriculture and Rural Development for their financial support of this work. The authors would like to thank Cannasoul Analytics and especially Dr. Rotem Perry Feigenbaum and Yogev Nir for the isolation of the phytocannabinoid compounds; Dr. Hugo Gottlieb and Dr. Michal Afri from the NMR unit at Bar Ilan University for performing the NMR measurements and deciphering the structure of the 331-18A phytocannabinoid; Moshe Barkaly for the semi-preparative HPLC fractionations of the whole extract; and Dr. Dafna Antes for the graphic scheme illustration. The authors declare no competing financial interests. Israel Cancer Research Fund grant 21-112-PG (DM). Gruss Lipper Charitable Foundation grant 2027093 (DM). Israeli Ministry of Agriculture and Rural Development 14370001 (DM).

## Additional information

### Competing interests

David Meiri: A patent pertaining to the results of the paper has been applied for the use of cannabinoids in the treatment of Notch-related diseases. This patent PCT/IL2021/051352 was licensed

to Cavnox Ltd. to test the combination of the molecules in human clinical trials. The other authors declare that no competing interests exist.

## Funding

| Funder | Grant reference number | Author |
|---|---|---|
| Israel Cancer Research Fund | 21-112-PG | David Meiri |
| Gruss-Lipper Family Foundation | 2027093 | David Meiri |
| Israeli Ministry of Agriculture and Rural Development | 14370001 | David Meiri |

The funders had no role in study design, data collection and interpretation, or the decision to submit the work for publication.

## Author contributions

Elazar Besser, Conceptualization, Formal analysis, Investigation, Visualization, Methodology, Writing – original draft, Writing – review and editing; Anat Gelfand, Formal analysis, Investigation; Shiri Procaccia, Conceptualization, Formal analysis, Visualization, Writing – original draft, Writing – review and editing; Paula Berman, Formal analysis, Investigation, Methodology, Writing – review and editing; David Meiri, Conceptualization, Supervision, Writing – original draft, Writing – review and editing

## Author ORCIDs

Elazar Besser (iD) https://orcid.org/0000-0002-5700-0947
David Meiri (iD) https://orcid.org/0000-0001-7627-1569

## Ethics

All procedures were performed according to protocols approved by the Technion Administrative Panel of Laboratory Animal Care (animal protocol number: IL_0470317).

Reviewer #2 (Public Review): https://doi.org/10.7554/eLife.90854.3.sa1
Author response https://doi.org/10.7554/eLife.90854.3.sa2

---

# Additional files

## Supplementary files

• Supplementary file 1. Phytocannabinoid concentrations by UHPLC/LC-MS of fractions relative to the whole extract.

• Supplementary file 2. Phytocannabinoid concentrations by UHPLC/LC-MS of Cannabis fraction 2 peaks.

• Supplementary file 3. $^1$H and $^{13}$C peak assignments and chemical shifts of 331-18A.

• Supplementary file 4. Ten most increased- and decreased-abundance genes following treatment of MOLT-4 cells with the whole extract according to Affymetrix.

• MDAR checklist

## Data availability

The sequencing data that support the findings of this study have been deposited in the Gene Expression Omnibus under the accession code GSE154287. All data generated or analyzed during this study are included in this published article and its supporting files.

The following dataset was generated:

| Author(s) | Year | Dataset title | Dataset URL | Database and Identifier |
|---|---|---|---|---|
| Meiri D, Besser E | 2023 | Expression data from MOLT-4 Cells treated with Cannabis extract | https://www.ncbi.nlm.nih.gov/geo/query/acc.cgi?acc=GSE154287 | NCBI Gene Expression Omnibus, GSE154287 |

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
