## [Editor Report · eLife assessment]

This **important** study follows up on previous work defining the anti-leukemic effects of a previously characterized cannabis extract on Notch-activated T cells and identifies several pathways that mediate its anti-cancer activity including the ER calcium and integrated stress response. The evidence is **solid**, but several concerns remain including the over reliance on a single cell line for the majority of the studies and lack of integration of the observations with existing literature

---

## [Referee Report · Reviewer #2 (Public Review)]

Summary:

The Meiri group previously showed that Notch1-activated human T-ALL cell lines are sensitive to a cannabis extract in vitro and in vivo (Ref. 32). In that article, the authors showed that Extract #12 reduced NICD expression and viability, which was partially rescued by restoring NICD expression. Here, the authors have identified three compounds of Extract #12 (CBD, 331-18A, and CBDV) that are responsible for the majority of anti-leukemic activity and NICD reduction. Using a pharmacological approach, the authors determined that Extract #12 exerted its anti-leukemic and NICD-reducing affects through the CB2 and TRPV1 receptors. To determine mechanism, the authors performed RNA-seq and observed that Extract #12 induces ER calcium depletion and stress-associated signals -- ATF4, CHOP, and CHAC1. Since CHAC1 was previously shown to be a Notch inhibitor in neural cells, the authors assume that the cannabis compounds repress Notch S1 cleavage through CHAC1 induction. The induction of stress-associated signals, Notch repression, and anti-leukemic effects were reversed by the integrated stress response (ISR) inhibitor ISRIB. Interestingly, combining the 3 cannabinoids gave synergistic anti-leukemic effects in vitro and had growth inhibitory effects in vivo.

Strengths:

(1) The authors show novel mechanistic insights that cannabinoids induce ER calcium release and that the subsequent integrated stress response represses activated NOTCH1 expression and kills T-ALL cells.

(2) This report adds to the evidence that phytocannabinoids can show a so-called "entourage effect" in which minor cannabinoids enhance the effect of the major cannabinoid CBD.

(3) This report dissects out the main cannabinoids in the previously described Extract #12 that contribute to T-ALL killing.

(4) The manuscript is clear and generally well-written.

(5) The data are mostly high quality and with adequate statistical analyses.

(6) The data generally support the authors' conclusions. The main exception is the experiments related to Notch.

(7) The authors' discovery of the role of the integrated stress response might explain previous observations that SERCA inhibitors block Notch S1 cleavage and activation in T-ALL (Roti Cancer Cell 2013). The previous explanation by Roti et al was that calcium depletion causes Notch misfolding, which leads to impaired trafficking and cleavage. Perhaps this explanation is not entirely sufficient?

Weaknesses:

(1) Given the authors' previous Cancer Communications paper on the anti-leukemic effects and mechanism of Extract #12, the significance of the original manuscript was reduced. To increase significance, the authors provided a new Fig. S7 in the revision showing that Extract #12 inhibits PDX growth in vivo. This experiment is nicely supportive of the anti-leukemic effects of Extract #12, raising the significance of their previous Cancer Communication paper by using in vivo patient-derived cells. However, this reviewer had suggested testing the combination of 333-18A+CBVD+CBD since the combination is the focus of the current manuscript. For unclear reasons, the combination was not tested.

(2) It would be important to connect the authors' findings and a wealth of literature on the role of ER calcium/stress on Notch cleavage, folding, trafficking, and activation. The several references suggested by this reviewer were not included in the revised manuscript for unclear reasons. These references are important to show the current status of the field and help readers appreciate what this manuscript brings that is new to T-ALL. In particular, Roti et al (Cancer Cell 2013) showed that SERCA inhibitors like thapsigardin reduce ER calcium levels and block Notch signaling by inhibiting NOTCH1 trafficking and inhibiting Furin-mediated (S1) cleavage of Notch1 in T-ALL. Multiple EGF repeats and all three Lin12/Notch repeats in the extracellular domains of Notch receptors require calcium for proper folding (Aster Biochemistry 1999; Gordon Nat. Struct. Mol. Biol. 2007; Hambleton Structure 2004; Rand Protein Sci 1997). Thus, Roti et al concluded that ER calcium depletion blocks NOTCH1 S1 cleavage in T-ALL. This effect seems to be conserved in *Drosophila* as Periz and Fortiin (EMBO J, 1999) showed impaired Notch cleavage in Ca2+/ATPase-mutated *Drosophila* cells.

(3) There is an overreliance of the data on single cell line -- MOLT4. MOLT4 is a good initial choice as it is Notch-mutated, Notch-dependent, and representative of the most common T-ALL subtype -- TAL1. However, there is no confirmatory data in other TAL1-positive T-ALLs or interrogation of other T-ALL subtypes. While this reviewer appreciates that the authors showed that multiple T-ALL cell lines were killed in response to Extract #12 in a previous study, the current manuscript is a separate study that should stand on its own. T-ALLs can be killed by multiple mechanisms. It would be important to show a few pieces of key data illustrating that the mechanism of killing found in MOLT4 applies to other T-ALLs.

(4) Fig. 6H. The effects of the cannabinoid combination might be statistically significant but seems biologically weak.

(5) Fig. 3. Based on these data, the authors conclude that the cannabinoid combination induces CHAC1, which represses Notch S1 cleavage in T-ALL cells. The concern is that Notch signaling is highly context dependent. CHAC1 might inhibit Notch in neural cells (Refs. 34-35), but it might not do this in a different context like T-ALL. It would be important to show evidence that CHAC1 represses S1 cleavage in the T-ALL context. More importantly, Fig. 3H clearly shows the cannabinoid combination inducing ATF4 and CHOP protein expression, but the effects on CHAC1 protein do not seem to be satisfactory as a mechanism for Notch inhibition. Perhaps something else is blocking Notch expression?

In the rebuttal, the two references provided by the authors do not alleviate concern that CHAC1 might not be acting as a Notch cleavage inhibitor in the T-ALL context. The Meng et al paper studied B-ALL not T-ALL and did not evaluate CHAC1 as a possible Notch cleavage inhibitor. Likewise, the Chang et al paper did not evaluate CHAC1 as a possible Notch cleavage inhibitor. Therefore, whether CHAC1 is a Notch cleavage inhibitor in the T-ALL context remains an open question. While the authors are correct that Supplementary Fig. S4G-I show that Extract #12 clearly induces CHAC1 protein expression, Main Fig. 3H shows that the extract combination 333-18A+CBVD+CBD, which is the focus of this manuscript, has unclear effects. If the extract combination has no effect on CHAC1 but has the same effects on Notch1 expression as the full extract, then there is reduced support for the authors' conclusion that the full extract and the 333-18A+CBVD+CBD combination inhibit Notch through CHAC1 induction.

(6) The authors provide a new figure on page 5 of the rebuttal that was not requested. It is supposed to show that CHAC1 loss protects T-ALL cells from Extract #12-induced cell population decline. Unfortunately, this figure is not conclusive. The empty vector PLKO is not an appropriate negative control. The field uses non-targeting shRNA controls like pLKO-luciferase to control for induction of the RNA interference pathway. Further, the viability data in panel B is normalized such that the effect of shCHAC1 on viability is masked. Showing non-normalized data is important, because if shCHAC1 impairs viability compared to control shRNA, then CHAC1might have effects on non-Notch pathways, which would reinforce the above concern in Point #5 that CHAC1 might not act as a Notch inhibitor in the T-ALL context. Separately, if this experiment had tested whether CHAC1 knockdown increases Notch cleavage and Notch target gene expression like DTX1, HES1 and MYC, then such data would have helped address Point #5.

(7) Fig. 4B-C/S5D-E. These Western blots of NICD expression are consistent with the cannabinoid combination blocking Furin-mediated NOTCH1 cleavage, which is reversed by ISR inhibition. However, there are many mechanisms that regulate NICD expression. To support their conclusion that the effects are specifically Furin-medated, the authors should probe full length (uncleaved) NOTCH1 in their Western blots. While the authors showed that the full extract (#12) increased uncleaved NOTCH1 expression in their Cancer Communications paper, a major conclusion of the manuscript is that the cannabinoid combination 333-18A+CBVD+CBD reproduces the effect of the full extract (#12). To support this conclusion, the authors should probe key blots for full-length Notch to show that the cannabinoid combination increases uncleaved NOTCH1 just like Extract #12 did in the authors' Cancer Communications paper.

(8) Fig. S4A-B. While these pharmacologic data are suggestive that Extract #12 reduces NICD expression through the CB2 receptor and TRPV1 channel, the doses used are very high (50uM). To exclude off-target effects, these data should be paired with genetic data to support the authors' conclusions. In the rebuttal, the authors provide dose response cell viability curves of the CB2 and TRPV1 inhibitors. These curves do not exclude the possibility that 50uM has off-target effects. This reviewer notes that Reviewer #1 had similar concerns and that both reviewers requested genetic validation of the pharmacological data. These data were not provided in the revision.

(9) Since the authors have performed gene expression profiling, an orthogonal test to confirm that Extract #12 acts through the Notch pathway is to perform enrichment analysis using Notch target gene signatures in T-ALL (e.g. Wang PNAS 2013). In contrast to the authors' rebuttal, this reviewer does not see any enrichment analysis (e.g. GSEA plots) performed on the microarray data to show that Extract #12 inhibits the Notch pathway.

(10) The revised manuscript still retains references that microarray data are "RNA-seq" data, which is inaccurate (see page 10, line 160; Figure 3 legend; page 12, line 169; page 27, line 428; page 36, line 741)

---

## [Author Response]

The following is the authors’ response to the original reviews.

**Reviewer #1 (Public Review):**
This is an interesting manuscript that extends prior work from this group identifying that a chemovar of Cannabis induces apoptosis of T-ALL cells by preventing NOTCH1 cleavage. Here the authors isolate specific components of the chemovar responsible for this effect to CBD and CBDV. They identify the mechanism of action of these agents as occurring via the integrated stress response. Overall the work is well performed but there are two lingering questions that would be helpful to address as follows:Exactly how CBD and CBDV result in the upregulation of the TRPV1/integrated stress response is unclear. What is the most proximal target of these agents that results in these changes?

The interaction of CBD and CBDV with TRPV1 has been thoroughly investigated by previous studies in the field. A few prominent examples are:

(1) De Petrocellis, Luciano, Alessia Ligresti, Aniello Schiano Moriello, Marco Allarà, Tiziana Bisogno, Stefania Petrosino, Colin G. Stott, and Vincenzo Di Marzo. "Effects of cannabinoids and cannabinoid‐enriched Cannabis extracts on TRP channels and endocannabinoid metabolic enzymes." British journal of pharmacology 163, no. 7 (2011): 1479-1494.

(2) Muller, Chanté, Paula Morales, and Patricia H. Reggio. "Cannabinoid ligands targeting TRP channels." Frontiers in molecular neuroscience 11 (2019): 487.

(3) Iannotti, Fabio Arturo, Charlotte L. Hill, Antonio Leo, Ahlam Alhusaini, Camille Soubrane, Enrico Mazzarella, Emilio Russo, Benjamin J. Whalley, Vincenzo Di Marzo, and Gary J. Stephens. "Nonpsychotropic plant cannabinoids, cannabidivarin (CBDV) and cannabidiol (CBD), activate and desensitize transient receptor potential vanilloid 1 (TRPV1) channels in vitro: potential for the treatment of neuronal hyperexcitability." ACS chemical neuroscience 5, no. 11 (2014): 1131-1141.

(4) Costa, Barbara, Gabriella Giagnoni, Chiara Franke, Anna Elisa Trovato, and Mariapia Colleoni. "Vanilloid TRPV1 receptor mediates the antihyperalgesic effect of the nonpsychoactive cannabinoid, cannabidiol, in a rat model of acute inflammation." British journal of pharmacology 143, no. 2 (2004): 247-250.

(5) de Almeida, Douglas L., and Lakshmi A. Devi. "Diversity of molecular targets and signaling pathways for CBD." Pharmacology research & perspectives 8, no. 6 (2020): e00682.

(6) Anand, Uma, Ben Jones, Yuri Korchev, Stephen R. Bloom, Barbara Pacchetti, Praveen Anand, and Mikael Hans Sodergren. "CBD effects on TRPV1 signaling pathways in cultured DRG neurons." Journal of Pain Research (2020): 22692278.

Similarly, other works have demonstrated the link between TRPV1 and the integrated stress response pathway (see below). These studies suggested increased reactive oxygen species (ROS) production, Cyclooxygenase-2 (COX-2) enzyme, as well as other stressors, lead to modulation of intracellular calcium levels by TRPV1.

(1) Ho, Karen W., Nicholas J. Ward, and David J. Calkins. "TRPV1: a stress response protein in the central nervous system." American journal of neurodegenerative disease 1, no. 1 (2012): 1.

(2) de la Harpe, Amy, Natasha Beukes, and Carminita L. Frost. "CBD activation of TRPV1 induces oxidative signaling and subsequent ER stress in breast cancer cell lines." Biotechnology and Applied Biochemistry 69, no. 2 (2022): 420-430.

(3) Soliman, Eman, and Rukiyah Van Dross. "Anandamide‐induced endoplasmic reticulum stress and apoptosis are mediated by oxidative stress in nonmelanoma skin cancer: Receptor‐independent endocannabinoid signaling." Molecular Carcinogenesis 55, no. 11 (2016): 1807-1821.

Related to the above, all experiments to confirm the mechanism of action of CBD/CBDV rely on chemical agents, whose precise targets are not fully clear in some cases. Thus, some use of genetic means (such as by knockout of TRPV1, ATF4) to confirm the dependency of these pathways on drug response and NOTCH cleavage would be very helpful.

Knockdown experiments and inhibition with inhibitors are two different approaches to studying the function of a specific gene or protein. Each method has its advantages and limitations. We initially attempted to knock-down CHAC1, but only managed to silence ~50% (Incomplete knockdown). Following treatment of MOLT4 cells with the whole extract, we observed only a partial downregulation in the mRNA expression of the Notch intracellular domain (NICD) (left panel), which could account for the incomplete rescue from the extract-induced death (right panel). We therefore turned to confirm the signaling pathway by inhibition of different targets with chemical agents.

**Author response image 1. sa2fig1:** Partial knockdown of CHAC1 hinders extract-induced cell death. (A) MOLT-4 cells were treated with either an empty vector or shRNA for Chac1, 369 and 739 represent two different areas of Chac1, for 48 hrs. Then, the gene expression of CHAC1 was assessed via qRT-PCR (N=3). (B) MOLT-4 cells were treated as in A, then added vehicle control or whole Extract (3 µg/mL) for additional 24 hrs, and the viability of the cells was assessed with XTT.

**Reviewer #2 (Public Review):**
Summary:The Meiri group previously showed that Notch1-activated human T-ALL cell lines are sensitive to a cannabis extract in vitro and in vivo (Ref. 32). In that article, the authors showed that Extract #12 reduced NICD expression and viability, which was partially rescued by restoring NICD expression. Here, the authors have identified three compounds of Extract #12 (CBD, 331-18A, and CBDV) that are responsible for the majority of anti-leukemic activity and NICD reduction. Using a pharmacological approach, the authors determined that Extract #12 exerted its anti-leukemic and NICD-reducing effects through the CB2 and TRPV1 receptors. To determine the mechanism, the authors performed RNA-seq and observed that Extract #12 induces ER calcium depletion and stress-associated signals -- ATF4, CHOP, and CHAC1. Since CHAC1 was previously shown to be a Notch inhibitor in neural cells, the authors assume that the cannabis compounds repress Notch S1 cleavage through CHAC1 induction. The induction of stress-associated signals, Notch repression, and anti-leukemic effects were reversed by the integrated stress response (ISR) inhibitor ISRIB. Interestingly, combining the 3 cannabinoids gave synergistic anti-leukemic effects in vitro and had growthinhibitory effects in vivo.Strengths:(1) The authors show novel mechanistic insights that cannabinoids induce ER calcium release and that the subsequent integrated stress response represses activated NOTCH1 expression and kills T-ALL cells.(2) This report adds to the evidence that phytocannabinoids can show a so-called "entourage effect" in which minor cannabinoids enhance the effect of the major cannabinoid CBD.(3) This report dissects the main cannabinoids in the previously described Extract #12 that contribute to T-ALL killing.(4) The manuscript is clear and generally well-written.(5) The data are generally high quality and with adequate statistical analyses.(6) The data generally support the authors' conclusions. The exception is the experiments related to Notch.(7) The authors' discovery of the role of the integrated stress response might explain previous observations that SERCA inhibitors block Notch S1 cleavage and activation in T-ALL (Roti Cancer Cell 2013). The previous explanation by Roti et al was that calcium depletion causes Notch misfolding, which leads to impaired trafficking and cleavage. Perhaps this explanation is not entirely sufficient.Weaknesses:(1) Given the authors' previous Cancer Communications paper on the anti-leukemic effects and mechanism of Extract #12, the significance of the current manuscript is reduced.

Our original manuscript consisted extensive multidisciplinary research, and we were asked to divide the research work into a paper that focuses on the cannabis plant and another paper that addresses finding the specific molecules and their underlying mechanism.

We understand that our publication of the initial observations with the whole extract dampened the overall novelty presented here, but the previous publication lacked the detailed and strong mechanistic work presented here that explains how the cannabis extract exerted its antitumoral effects.

In addition, the finding of the need for 3 phytocannabinoids and the synergy analysis supplies essential support to the ‘entourage effect’. This is a phenomenon in which the presence of minor proportions of cannabinoids and other plant components significantly modulate the effects of the main active components of cannabis and thereby produce more potent or more selective effects than the use of one major cannabinoid alone. It was well-demonstrated for endocannabinoids but was only demonstrated in very few studies for phytocannabinoids.

(2) It would be important to connect the authors' findings and a wealth of literature on the role of ER calcium/stress on Notch cleavage, folding, trafficking, and activation.

We mentioned three previous papers (ref. 34-36) that guided us in our investigation. Following this reviewer’s comment, we added to the discussion a few lines connecting our findings to previous works on ER stress and Notch activation with the appropriate references.

(3) There is an overreliance on the data on a single cell line -- MOLT4. MOLT4 is a good initial choice as it is Notch-mutated, Notch-dependent, and representative of the most common T-ALL subtype -- TAL1. However, there is no confirmatory data in other TAL1positive T-ALLs or interrogation of other T-ALL subtypes.

As mentioned by the reviewer, this study followed a previous publication in which 7 different cell lines were assessed (MOLT‐4, CCRF‐CEM, Jurkat, Loucy, HPB-ALL, DND-41and T-ALL1). MOLT-4 cells were used to investigate the mechanism, both MOLT-4 cells and CCRF-CEM cells were utilized to investigate the effect of the cannabinoid combination or the whole extract in-vivo.

(4) Fig. 6H. The effects of the cannabinoid combination might be statistically significant but seem biologically weak.

Survival rates are presented in Fig. 6H for the combination of the cannabinoids and in Supplementary Fig. S6C for the whole extract. While this mouse model provides valuable insights, the biological significance and the translation of findings to human patients require cautious interpretation.

(5) Fig. 3. Based on these data, the authors conclude that the cannabinoid combination induces CHAC1, which represses Notch S1 cleavage in T-ALL cells. The concern is that Notch signaling is highly context-dependent. CHAC1 might inhibit Notch in neural cells (Refs. 34-35), but it might not do this in a different context like T-ALL. It would be important to show evidence that CHAC1 represses S1 cleavage in the T-ALL context. More importantly, Fig. 3H clearly shows the cannabinoid combination inducing ATF4 and CHOP protein expression, but the effects on CHAC1 protein do not seem to be satisfactory as a mechanism for Notch inhibition. Perhaps something else is blocking Notch expression?

We understand the reviewer’s concern. Previous works had shown the upregulation of CHAC1 also in the context of Notch signaling in leukemia, particularly recently also for T-ALL:

(1) Meng, X., Matlawska-Wasowska, K., Girodon, F., Mazel, T., Willman, C.L., Atlas, S., Chen, I.M., Harvey, R.C., Hunger, S.P., Ness, S.A. and Winter, S.S., 2011. GSI-I (Z-LLNle-CHO) inhibits γ-secretase and the proteosome to trigger cell death in precursor-B acute lymphoblastic leukemia. Leukemia, 25(7), pp.11351146.

(2) Chang, Yoon Soo, Joell J. Gills, Shigeru Kawabata, Masahiro Onozawa, Giusy Della Gatta, Adolfo A. Ferrando, Peter D. Aplan, and Phillip A. Dennis. "Inhibition of the NOTCH and mTOR pathways by nelfinavir as a novel treatment for T cell acute lymphoblastic leukemia." International Journal of Oncology 63, no. 5 (2023): 1-12.

As for the second part of the reviewer’s comment, we tested both the mRNA transcript and protein expression of CHAC1. The increase is clearly shown at 60 min for the mRNA Fig. 3D and Fig. 4F and for the protein also in Supplementary Fig. S4G-I.

To show direct involvement of CHAC1 we utilized the means of knockdown. Though it was not completely effective and accounted for about ~50% reduction, it clearly shows the involvement of CHAC1 in the mechanism leading to the reduction in viability of these cancer cells.

**Author response image 2. sa2fig2:** Partial knockdown of CHAC1 hinders extract-induced cell death. (A) MOLT-4 cells were treated with either an empty vector or shRNA for Chac1, 369 and 739 represent two different areas of Chac1, for 48 hrs. Then, the gene expression of CHAC1 was assessed via qRT-PCR (N=3). (B) MOLT-4 cells were treated as in A, then added vehicle control or whole Extract (3 µg/mL) for additional 24 hrs, and the viability of the cells was assessed with XTT.

(6) Fig. 4B-C/S5D-E. These Western blots of NICD expression are consistent with the cannabinoid combination blocking Furin-mediated NOTCH1 cleavage, which is reversed by ISR inhibition. However, there are many mechanisms that regulate NICD expression. To support their conclusion that the effects are specifically Furin-medated, the authors should probe full-length (uncleaved) NOTCH1 in their Western blots.

We have probed for the full-length Notch1 in our previously published paper (Cancer Communications, Supplementary Fig. S1G-I). As we have shown here the three cannabinoids together mimic the effect of the whole extract, we did not repeat the experiments with full-length Notch1.

(7) Fig. S4A-B. While these pharmacologic data are suggestive that Extract #12 reduces NICD expression through the CB2 receptor and TRPV1 channel, the doses used are very high (50uM). To exclude off-target effects, these data should be paired with genetic data to support the authors' conclusions.

We performed a dose-response experiment before choosing the doses used for the inhibitors of CB2 and TRPV1 (see below). The dose of 50 µM was selected as it did not affect the viability of the cells.

**Author response image 3. sa2fig3:** Dose-response of CB2 and TRPV1 inhibitors in MOLT-4 cells. MOLT-4 cells were treated with increasing concentrations (µM) of (A) CB2 inhibitor AM630 or (B) TRPV1 inhibitor AMG9810; and 24 hrs later the viability of the cells was assessed with XTT.

**Reviewer #2 (Recommendations For The Authors):**
(1) In Fig. 6H, it is unclear why the authors are using CCRF-CEM cells, which are known to be resistant to Notch inhibitors, rather than popular cell lines that are Notch-dependent (e.g. CUTLL1, DND-41, HPB-ALL). Since cannabinoids seem to kill at least in part through Notch inhibition, the effects would be predicted to be greater in Notch-dependent T-ALL cell lines than Notch-independent cell lines like CCRF-CEM. To show stronger in vivo preclinical efficacy, another suggestion is to combine cannabinoids with tolerable dosing of gammasecretase inhibitors as published by the Michelle Kelliher group.

We have shown in our previous publication that both MOLT-4 and CCRF-CEM cells are dependent on Notch for their propagation, while other cell lines of T-ALL such as Loucy and Jurkat do not. Therefore, we treat CCRF-CEM as Notch-dependent. We discuss the possibility of using the cannabinoid combination with other treatments, specifically chemotherapy, to enhance effectiveness.

(2) To increase significance, this reviewer suggests strengthening the mechanism. However, this reviewer understands the challenge of identifying the correct mechanism. Thus, an alternative would be to increase clinical relevance. Some specific suggestions are described below.(a) With regard to increasing mechanistic insights, the authors should be aware of some papers that might be helpful. Roti et al (Cancer Cell 2013) showed that SERCA inhibitors like thapsigardin reduce ER calcium levels and block Notch signaling by inhibiting NOTCH1 trafficking and inhibiting Furin-mediated (S1) cleavage of Notch1. Multiple EGF repeats and all three Lin12/Notch repeats in the extracellular domains of Notch receptors require calcium for proper folding (Aster Biochemistry 1999; Gordon Nat. Struct. Mol. Biol. 2007; Hambleton Structure 2004; Rand Protein Sci 1997). Thus, Roti et al concluded that ER calcium depletion blocks NOTCH1 S1 cleavage. This effect seems to be conserved in *Drosophila* as Periz and Fortiin (EMBO J, 1999) showed impaired Notch cleavage in Ca2+/ATPasemutated *Drosophila* cells. Besser et al should consider these papers when exploring the mechanism by which the ER calcium release by the cannabinoid combination blocks activated NOTCH1 expression. Similarities and differences should be discussed.

As mentioned above and stated also by the reviewer, many papers have shown the cleavage and/or activation of Notch following ER stress.

(b) With regard to increasing clinical relevance, the authors should consider testing the effects of the cannabinoid combination on primary samples, PDX models, and/or genetically engineered mouse models. Pan-Notch inhibitors like gamma-secretase inhibitors (GSIs) have been disappointing in clinical trials because of excessive on-target toxicity, in particular in the intestine. The authors should consider exploring whether the cannabinoids might be superior to GSIs with regard to intestinal toxicity and why that might be (e.g. receptor expression).

We thank the reviewer and agree that clinical relevance is of outmost importance. As obtaining primary tumor cells from patients is challenging, we assessed the whole cannabis extract in a PDX model. This extract is already being used by patients. We added this result as Supplementary fig. S7, and address it in the main text of the Results and in the Materials and Methods section.

(3) Since the authors have performed gene expression profiling, another test to confirm that Extract #12 acts through the Notch pathway is to perform enrichment analysis for known Notch target genes in T-ALL (e.g. Wang PNAS 2013).

We performed the analysis and this is how we pinpointed the involvement of ATF4, CHOP and CHAC1 of the integrated stress response pathway.

Minor concern:Supplemental Table S4. According to the text (page 10, line 160) and table title, these data are RNA-seq. However, according to the GSE154287 annotation, these data are Affymetrix arrays There are no gene names in the GSE table. Are the IDs probesets rather than genes?

Indeed, the gene analysis data are Affymetrix arrays and the title was corrected.